# Minimal impact of recent decline in C$_4$ vegetation abundance on atmospheric carbon isotopic composition

Aliénor Lavergne [1,6] ✉, Sandy P. Harrison [1,2], Kamolphat Atsawawaranunt [1,3], Ning Dong [4,5] & Iain Colin Prentice [2,5] ✉

Changes in atmospheric carbon dioxide concentrations, climate, and land management influence the abundance and distribution of C$_3$ and C$_4$ plants, yet their impact on the global carbon cycle remains uncertain. Here, we use a parsimonious model of C$_3$ and C$_4$ plant distribution, based on optimality principles, combined with a simplified representation of the global carbon cycle, to assess how shifts in plant abundances driven by carbon dioxide and climate affect global gross primary production, land carbon isotope discrimination, and the isotopic composition of atmospheric carbon dioxide. We estimate that the proportion of C$_4$ plants in total biomass declined from about 16% to 12% between 1982 and 2016, despite an increase in the abundance of C$_4$ crops. This decline reflects the reduced competitive advantage of C$_4$ photosynthesis in a carbon dioxide-enriched atmosphere. As a result, global gross primary production rose by approximately 16.5 ± 1.8 petagrams of carbon, and land carbon isotope discrimination increased by 0.017 ± 0.001‰ per year. Accounting for changes in C$_3$ and C$_4$ abundances reduces the difference between observed and modeled trends in atmospheric carbon isotope composition, but does not fully explain the observed decrease, pointing to additional, unaccounted drivers.

The accumulation of carbon dioxide (CO$_2$) in the atmosphere due to fossil fuel burning has increased global gross primary production (GPP) and decreased the isotopic composition of atmospheric CO$_2$ ($\delta^{13}CO_2$) – the ratio of the heavier ($^{13}C$) to the lighter ($^{12}C$) stable carbon isotope of atmospheric CO$_2$ – over the past century, a phenomenon known as the Suess effect[1]. While changes in GPP can influence carbon isotope discrimination ($\Delta^{13}C$) during photosynthesis, the relationship is not strictly linear and depends on environmental and physiological conditions[2], which means shifts in plant productivity may affect $\delta^{13}CO_2$ in complex ways. Atmospheric measurements show that $\delta^{13}CO_2$, expressed as the normalized ratio of $^{13}C$ to $^{12}C$ compared to a standard, has declined by 0.027‰ per year over the period from 1978 to 2014[3,4]. However, the observed $\delta^{13}CO_2$ decrease is apparently smaller than expected when accounting for land and ocean carbon cycling and uptake in a simple calibrated model, which predicts a decline of about 0.032‰ per year[3]. To explain this shortfall, $\Delta^{13}C$ of the terrestrial biosphere should have increased by 0.014 ± 0.007‰ per ppm of CO$_2$ increase[3]. Whether a $\Delta^{13}C$ change of this magnitude is consistent with actual terrestrial carbon fluxes remains uncertain.

Global $\Delta^{13}C$ estimates from long-term tree-ring measurements do not show the increase in the $\Delta^{13}C$ of C$_3$ plants postulated by ref. 3. Although the $\Delta^{13}C$ of C$_3$ plants as recorded by tree rings has been variable across sites (increasing, decreasing, or unchanging[2,5,6]), globally it has remained roughly constant[2]. It is possible that post-photosynthetic fractionation processes[2,7] and intrinsic age-related changes in tree development over their lifespan, such as tree height[8,9] could affect inferences of long-term $\Delta^{13}C$ trends from tree rings. However, these effects are not well understood or quantified. No alternative source of data for C$_3$ plants are currently available. The mean residence time of C$_4$ plant-derived carbon in the biosphere is generally shorter than that of C$_3$ plant-derived carbon[10,11] and there is no direct evidence of changes in $\Delta^{13}C$ of C$_4$ plants. Since atmospheric measurements reflect large-scale changes in vegetation dynamics while in situ measurements only record ecophysiological adjustments from the ecosystems studied, it is challenging to reconcile measurements on the ground with estimates from the atmosphere. Current models linking carbon fluxes and stocks between land and the atmosphere are complex, making comparison difficult. Simple modeling approaches are needed to determine recent

[1]Department of Geography and Environmental Science, University of Reading, Berkshire, UK. [2]Ministry of Education Key Laboratory for Earth System Modeling, Department of Earth System Science, Tsinghua University, Beijing, China. [3]Department of Biological Sciences, City Campus, Auckland University, Auckland, New Zealand. [4]College of Resources and Environment, Huazhong Agricultural University, Wuhan 430070 China. [5]Georgina Mace Centre for the Living Planet, Department of Life Sciences, Imperial College London, Silwood Park Campus, Ascot, UK. [6]Present address: Nature Geoscience, Springer Nature Group, The Campus, London, UK. ✉e-mail: alienor.lavergne@gmail.com; c.prentice@imperial.ac.uk

changes in global GPP and associated $\Delta^{13}C$, and to disentangle the contribution from $C_3$ and $C_4$ plants to the observed atmospheric trends.

The global distribution of $C_3$ and $C_4$ plants reflects their divergent responses to climate as well as human activities via cropping and land management[11,12]. $C_3$ plants include cool-climate grasses, most shrubs, and nearly all trees[13], while $C_4$ plants generally dominate in warm-climate grasslands and savannas. $C_4$ plants possess a unique set of adaptations making them more competitive than $C_3$ plants in warm, arid, and high-light environments[14,15], primarily via reduced rates of photorespiration. In contrast, $C_3$ photosynthesis is stimulated at high atmospheric $CO_2$ concentrations. This is known as the $CO_2$ fertilization effect and confers an advantage over $C_4$ photosynthesis under elevated $CO_2$[16].

Variations in $\Delta^{13}C$ are closely related to environmentally driven changes in the stomatal limitation of photosynthesis, expressed as the ratio of leaf-internal to ambient partial pressures of $CO_2$. $\Delta^{13}C$ also depends on the pathway of carbon assimilation. Isotopic fractionation during the diffusion of $CO_2$ through the stomata primarily influences $\Delta^{13}C$ in $C_4$ plants, while fractionation during Rubisco carboxylation has a stronger imprint on $\Delta^{13}C$ in $C_3$ plants, resulting in $C_3$ plants being depleted in $^{13}C$ compared to $C_4$ plants[17–19]. Knowledge of the different isotopic signatures of $C_3$ and $C_4$ photosynthetic pathways and of their relative coverage across the globe can be used to estimate average $\delta^{13}C$ across terrestrial environments, and so global $\Delta^{13}C$.

Several models of the distribution of $C_3$ and $C_4$ plants have been proposed. By far the most widely used $C_4$ distribution map in ecophysiological research and land-surface modeling is the one developed by ref. 20 based on an approach published in ref. 11—see for example,[21–24]. However, this map is static, implying constancy over time. More recent work has considered the differential responses of $C_3$ and $C_4$ plants to recent environmental changes, based on an optimality model[12]. The derived map[12] indicates that the global fraction of $C_4$ plants has decreased over 2001–2019 due to a decrease in the natural abundance of $C_4$ grasses with elevated $CO_2$, even as the area of $C_4$ crops increased. Such a shift would impact trends in global GPP and $\Delta^{13}C$. Reference 2 suggested that the lower-than-expected decrease in global $\delta^{13}CO_2$ observed in atmospheric measurements (attenuation of the Suess effect)[3] might be explained by changes in the abundance and distribution of $C_3$ and $C_4$ plants. In contrast, ref. 3 suggested that any change in $C_3/C_4$ distribution would have a negligible impact on atmospheric $\delta^{13}CO_2$, given the higher carbon turnover rate in $C_4$ than $C_3$ plants—implying a dominant control on atmospheric $\delta^{13}CO_2$ by $C_3$ photosynthesis.

Here we propose a new $C_3/C_4$ distribution model based on the well validated, optimality-based P model[25–27] to test refs. 2, 3 hypotheses (hereafter denoted as Lavergne2022 and Keeling2017, respectively) and to determine whether changes in $C_3$ and $C_4$ plant distributions could explain the observed decrease in atmospheric $\delta^{13}CO_2$ over the period from 1982 to 2016 (see workflow in Supplementary Fig. 1). We compiled a large global dataset of stable carbon isotope measurements from leaves[28] and soils[29] to evaluate model predictions of photosynthetic $\Delta^{13}C$ in $C_3$ and $C_4$ plants and $C_3/C_4$ fractions, respectively. $\delta^{13}C_{soil}$ is a good indicator of local changes in

the abundance of $C_3$ and $C_4$ plants because of the contrasting isotopic signatures of the two photosynthetic pathways. We estimated recent changes in the abundance and distribution of $C_3$ and $C_4$ plants, GPP and $\Delta^{13}C$ in response to environmental changes and compared them with those based on the $C_4$ distribution maps of ref. 20 and ref. 12 (hereafter denoted as Still2009 and Luo2014, respectively). We performed an attribution analysis to determine the relative contributions of environmental drivers to the changes in the fraction of $C_4$ plants, GPP, and $\Delta^{13}C$. Finally, we used a simple carbon-cycle box model[30,31] to determine whether changes in $C_3$ and $C_4$ biomass, weighted by their relative fractions and carbon turnover times, could explain the magnitude of the observed decrease in atmospheric $\delta^{13}CO_2$.

## Results

The skill of the model to predict $\Delta^{13}C$ for $C_3$ and $C_4$ plants was good with coefficients of determination ($R^2$) averaging 0.50, 0.23 and 0.92, respectively for $C_3$, $C_4$ and total ($C_3$ and $C_4$) plants (Fig. 1a). However, the model underestimated the leaf-derived variability of $\Delta^{13}C$ (standard deviation = 1.56‰ versus 2.63‰ for $C_3$ plants, and 0.60‰ versus 1.32‰ for $C_4$ plants, respectively for model and observations). The model reproduced 58% of the variability of the global soil isotopic $\delta^{13}C_{soil}$ records ($R^2 = 0.58$; Fig. 1b), an improvement over the simulations using the $C_4$ maps of Still2009 and Luo 2024 ($R^2 = 0.32$ and 0.37, respectively; Fig. 1c, d).

The predicted fraction of $C_4$ plants ($F_4$) was large in hot and dry regions including subtropical Africa, the southern part of North America, northeastern Brazil and northern Australia, but low in cold and temperate regions (Fig. 2a). Similar patterns of spatial variation were shown in Still2009 and Luo2024 (Fig. 2b, c), but were more pronounced in Still2009 than in the other two maps. Predicted $F_4$ tended to decrease over the period studied in most regions with $F_4 > 5\%$ (Fig. 3a), but to increase slightly in southern equatorial regions (0–20°S) in central Eurasia and high latitudes of North America (Fig. 3b). The predicted $F_4$ decrease was greater than in Luo2024 over their common 2001–2016 period (Fig. 3b, c). Luo2024 showed increases in $F_4$ in more regions than in our map (Fig. 3c). Globally, according to our model, $F_4$ decreased from 14.1 to 10.3% for natural grasslands but increased from 1.7 to 2.0% for crops between 1982 and 2016. This resulted in a decrease of global $F_4$ (considering both natural grasslands and crops) from 15.8 to 12.2% over the same period (Fig. 4a), and from 13.6 to 12.2%. Global $F_4$ was 13.8% in Still2009 but decreased over 2001–2016 from 12.5 to 12.3% in Luo2024 (Fig. 4b). $F_4$ decreased by 0.001% $yr^{-1}$ (p < 0.001) in our analysis, but at a slower rate for Luo2024 (<0.001% $yr^{-1}$, p < 0.01). The global average $F_4$ over the common 2001–2016 period was 12.7% using our model, similar to Luo2024 (12.5%).

From 1982 to 2016, while the predicted gross primary production (GPP) per unit land area for $C_3$ plants ($GPP_{C3}$) increased across the globe, GPP for $C_4$ plants ($GPP_{C4}$) decreased in line with the $F_4$ decrease (Supplementary Fig. 2a, b). Total GPP, including both $C_3$ and $C_4$ photosynthesis, increased almost everywhere, but decreased in South America around 30°S, in northwestern Australia and in Africa around 20°N and 20°S

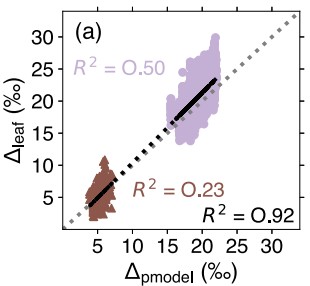 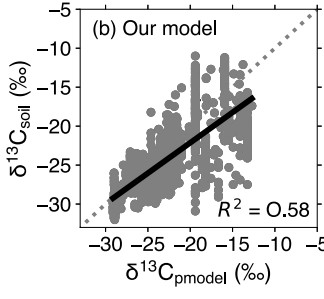 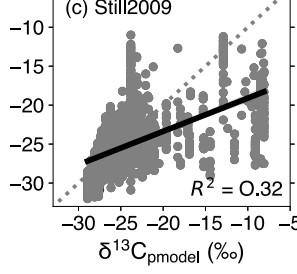 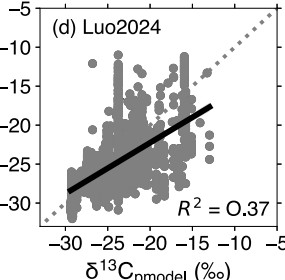

**Fig. 1 | Predictive skill of the model to reproduce leaf and soil stable carbon isotope data. a** Comparison between predicted and observed $\Delta^{13}C$ for $C_3$ (purple) and $C_4$ (brown) plants with associated coefficients of determination ($R^2$). The $R^2$ for all ($C_3 + C_4$) plants is shown in black. Comparison between predicted and observed $\delta^{13}C_{soil}$ for **b** our $C_3/C_4$ competition map, **c** the global $C_4$ map of Still2009, and **d** Luo2024, with associated coefficients of determination ($R^2$).

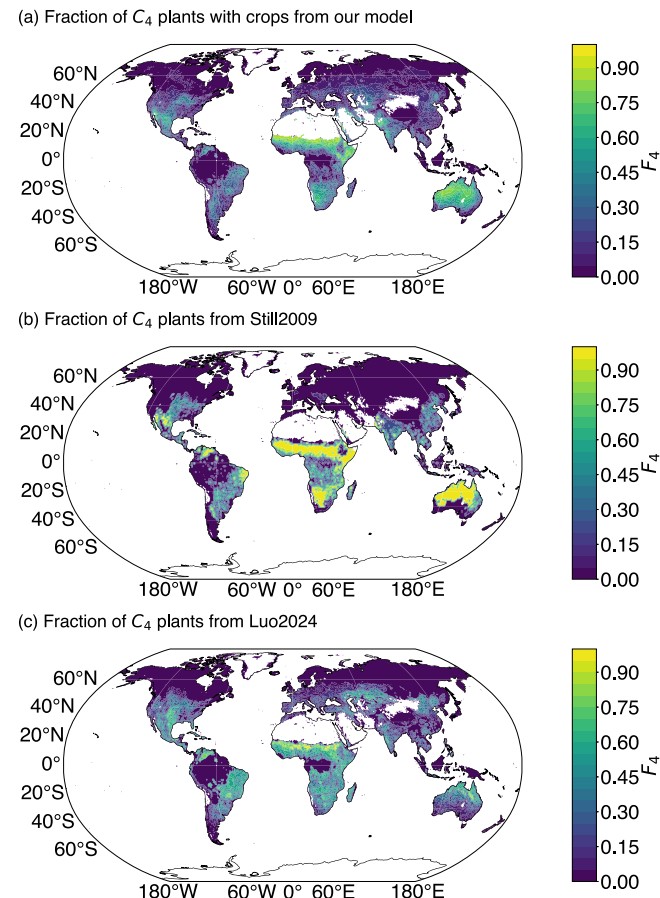

Fig. 2 | **Predicted fraction of C4 plants across the globe over 2001–2016.** Model predictions using our simple $C_3/C_4$ approach, including both natural grasslands and croplands (**a**). Global $C_4$ distribution maps from Still2009 (**b**) and Luo2024 (**c**).

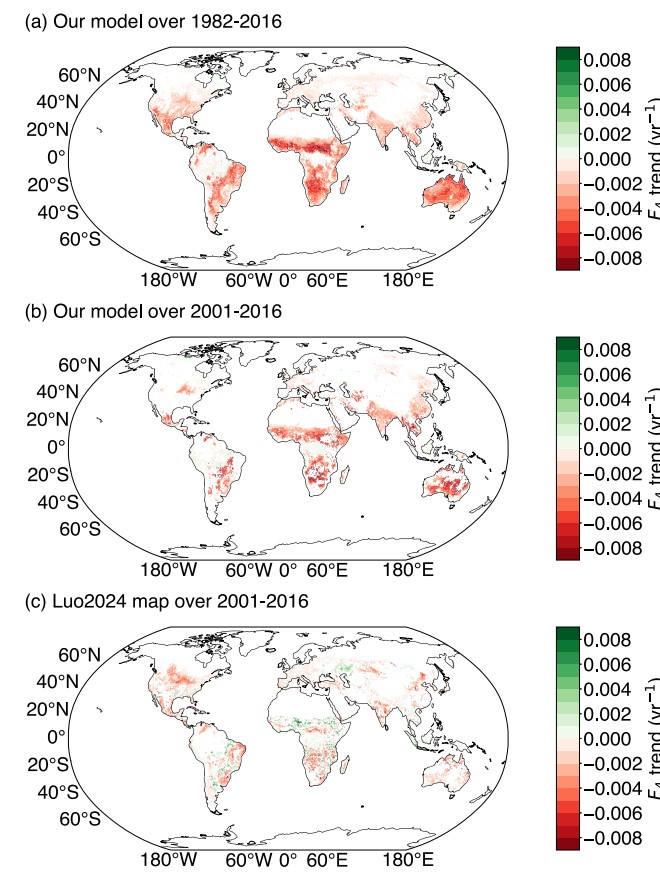

Fig. 3 | **Temporal trend in the global fraction of C4 plants ($F_4$, in yr$^{-1}$).** Trend in $F_4$ from our $C_3/C_4$ map over 1982–2016 (**a**) and 2001–2016 (**b**). Trend in $F_4$ from Luo over 2001–2016 (**c**).

(Supplementary Fig. 2c). Globally, GPP$_{C3}$ increased at a rate of $0.75 \pm 0.06$ PgC yr$^{-2}$ and GPP$_{C4}$ decreased by $0.28 \pm 0.02$ PgC yr$^{-2}$ (Fig. 4c), resulting in an increase of total GPP (including $C_3$ and $C_4$ natural grasslands and crops) by $0.47 \pm 0.05$ PgC yr$^{-2}$—equivalent to an increase of $16.5 \pm 1.8$ PgC yr$^{-1}$— during 1982–2016 (Fig. 4d). These estimates differ from those obtained using the fixed $C_4$ distribution map of Still2009, where GPP increased at a faster rate of $0.53 \pm 0.06$ PgC yr$^{-2}$—equivalent to an increase of $18.6 \pm 2.1$ PgC—with GPP$_{C3}$ and GPP$_{C4}$ increasing by $0.48 \pm 0.04$ PgC yr$^{-2}$ and $0.07 \pm 0.02$ PgC yr$^{-2}$ respectively (Fig. 4d). Over the 2001–2016 period, the rate of increase in GPP was slightly lower using our map than that of Still2009 or Luo2024: $0.45 \pm 0.11$ compared to $0.46 \pm 0.12$ and $0.50 \pm 0.13$ PgC yr$^{-2}$, respectively. The mean GPP was similar using our map and that of Luo2024 (Fig. 4d). The average contribution of $C_4$ photosynthesis to global GPP over 2001–2016 was around 16.2% for natural grasslands, and 18.5, 18.6, and 17.5% for total $C_4$ plants (including crops) based on our map and the Still2009 and Luo2024 maps, respectively.

Predicted $\Delta^{13}C$ increased for $C_3$ plants ($\Delta^{13}C_{C3}$) but decreased for $C_4$ plants ($\Delta^{13}C_{C4}$) from 1982 to 2016 (Supplementary Fig. 3a, b), resulting in an increase of total $\Delta^{13}C$, including both $C_3$ and $C_4$ photosynthesis, almost everywhere (Supplementary Fig. 3c). Globally, when using a constant $F_4$ from the Still2009 map, $\Delta^{13}C$ increased by $0.005 \pm 0.001$‰ yr$^{-1}$ for $C_3$ plants over 1982–2016 ($p < 0.001$) equivalent to $0.003 \pm 0.001$‰ per ppm increase of $CO_2$, while $\Delta^{13}C$ for $C_4$ plants stayed broadly constant, decreasing by $<0.001$‰ yr$^{-1}$ ($p < 0.001$) equivalent to $<0.001$‰ ppm$^{-1}$ (Fig. 4e), resulting in a total $\Delta^{13}C$ (including both $C_3$ and $C_4$ plants) increase of $0.005 \pm 0.001$‰ yr$^{-1}$ (Fig. 4f) equivalent to $0.003 \pm 0.001$‰ ppm$^{-1}$. However, when considering the global decrease in $F_4$ over 1982–2016 using our $C_4$ map, $\Delta^{13}C$ in $C_3$ plants increased at a faster rate ($0.018 \pm 0.001$‰ yr$^{-1}$, equivalent to $0.010 \pm 0.001$‰ ppm$^{-1}$, $p < 0.001$), while $\Delta^{13}C$ in $C_4$ plants

decreased by $0.003 \pm 0.001$‰ yr$^{-1}$, equivalent to $0.002 \pm 0.001$‰ ppm$^{-1}$ ($p < 0.001$), leading to a total $\Delta^{13}C$ increase of $0.017 \pm 0.001$‰ yr$^{-1}$, equivalent to $0.010 \pm 0.001$‰ ppm$^{-1}$ ($p < 0.001$). Over the common period 2001–2016, global $\Delta^{13}C$ increased by $0.004 \pm 0.002$‰ yr$^{-1}$, equivalent to $0.002 \pm 0.001$‰ ppm$^{-1}$ ($p < 0.01$) using Luo2024, and $0.014 \pm 0.005$‰ yr$^{-1}$, equivalent to $0.007 \pm 0.001$‰ ppm$^{-1}$, using our map.

The decrease in global $F_4$ was mainly driven by the increase in atmospheric $CO_2$ concentration ($c_a$), followed by increasing daytime air temperature ($T_{air}$) and, to a lesser extent, by increasing daytime vapor pressure deficit (VPD) and soil moisture ($\theta$) (Fig. 5a, b). Compared to $c_a$, $T_{air}$ and VPD, which showed spatially more-or-less homogeneous impacts on $F_4$, the $\theta$ contribution to $F_4$ was heterogeneous across the globe (Supplementary Fig. 4), reflecting heterogeneity in hydroclimatic changes (Supplementary Fig. 5). As expected, rising $c_a$ was the major contributor of GPP increase for $C_3$ plants, while rising $T_{air}$ increased GPP in $C_4$ plants (Fig. 5c–e). $T_{air}$ was the most important driver of $\Delta^{13}C$ for both $C_3$ and $C_4$ plants, followed by VPD. $\Delta^{13}C$ in $C_3$ plants increased most with higher $T_{air}$, followed by $c_a$ and to a lesser extent $\theta$, but decreased with higher VPD (Fig. 5f, g). In contrast, the decrease in $\Delta^{13}C$ for $C_4$ plants was mainly driven by rising $T_{air}$, while the increase in VPD attenuated this decrease (Fig. 5f, h).

Running the simple carbon cycle box model, in its original version[30], we were able to reproduce the results of ref. 3: a larger-than-observed decrease in atmospheric $\delta^{13}CO_2$ when assuming a constant $\Delta^{13}C$ (18‰), but consistent observed and predicted trends when using a varying $\Delta^{13}C$ driven by increases in atmospheric $CO_2$ (Fig. 6a). Assuming a constant $F_4$, and with the simplifying assumption that box 1 of the model (with small biomass and fast turnover) represents $C_4$ vegetation while boxes 2 and 3 (with intermediate to high biomass and intermediate to low turnover) include only $C_3$ vegetation, we predicted a faster-than-observed decrease in atmospheric $\delta^{13}CO_2$ (Fig. 6b). The difference between observations and predictions was

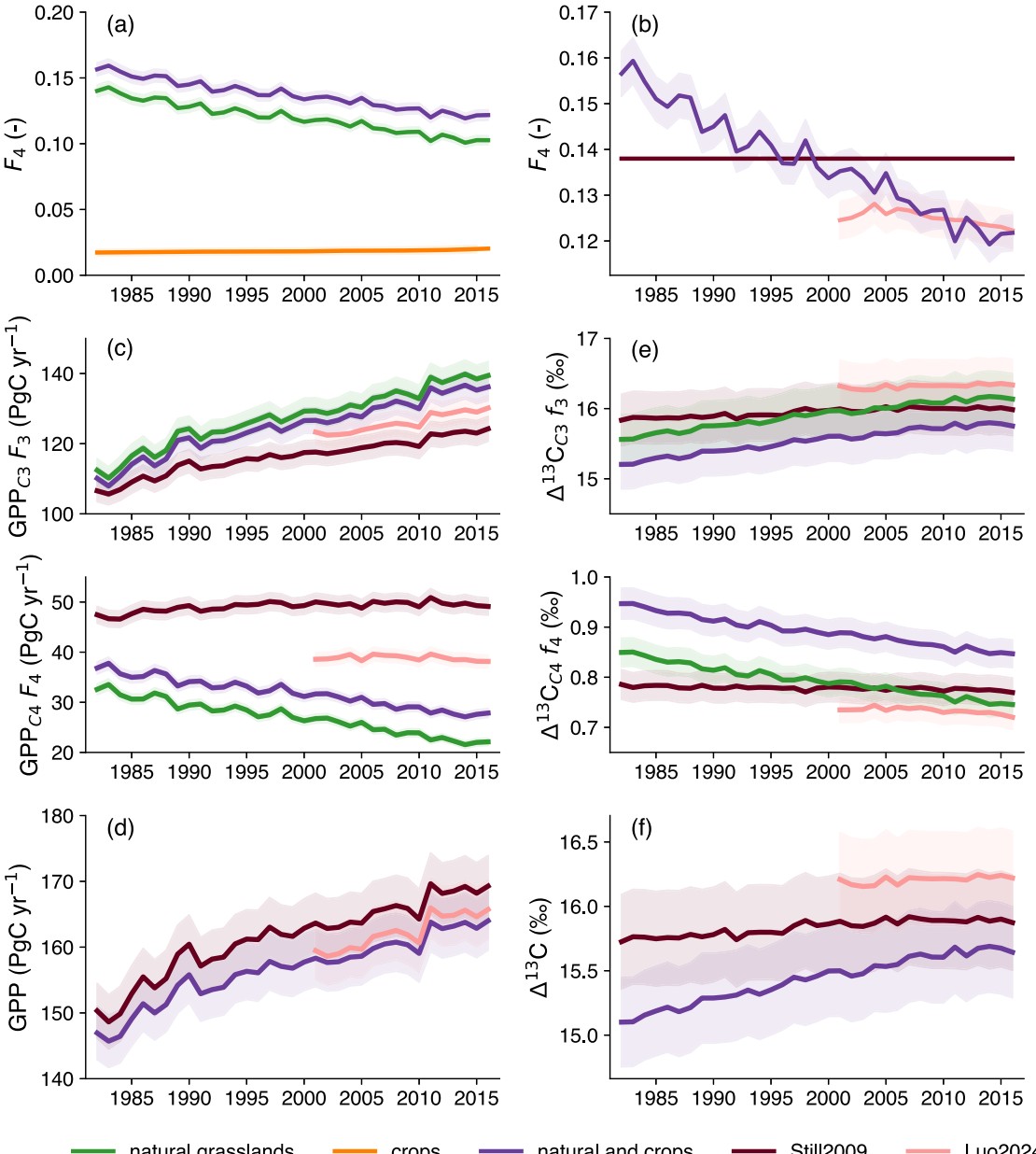

**Fig. 4 | Global changes in the fraction of C₄ plants, GPP and Δ¹³C over 1982–2016.** Fraction of C₄ plants ($F_4$) from our model for (**a**) natural grasslands (green), crops (orange) and both natural grasses and crops (violet) and (**b**) total $F_4$ compared to that from Still2009 (brown) and Luo2024 (light pink). GPP weighted by the fraction of C₃/C₄ plants (in PgC yr⁻¹) for natural grasslands (green), crops (orange) and both natural grasses and crops (violet) in C₃ (**c**) and C₄ (**d**) plants and (**e**) for all plants (including C₃ and C₄ plants; Eq. 3c) using our maps and those. Δ¹³C weighted by the fraction of C₃/C₄ plants (in ‰) for natural grasslands (green), crops (orange) and both natural grasses and crops (violet) in C₃ (**f**) and C₄ (**g**) plants and (**h**) for all plants (including C₃ and C₄ plants; Eq. 4) using our map, Still2009 and Luo2024. The light shade is the uncertainty calculated as the 95% confidence interval of the simulated mean (**a**, **b**, **f**, **g**, **h**) or sum (**c**–**e**) plus the sum of uncertainties from the input data (assumed to be ±2% of the global average values). The uncertainty range of the C₄ cropland area derived from LUHv2-2019 (orange) is assumed to be ±10% of the global average values following Luo.

greater when we assumed a constant Δ¹³C (equal to 6‰ for C₄ plants and 18‰ for C₃ plants) than when we let Δ¹³C vary annually using our global model outputs (difference between observed and predicted Δ¹³C of around 0.30 versus 0.16‰ in 2012–2016). When accounting for both varying Δ¹³C and $F_4$, our predicted δ¹³CO₂ values were closer to the observed trends, but they did not fully reproduce the magnitude of the decrease (the difference was around 0.08‰ over 2012–2016). Thus, by accounting for both changes in Δ¹³C and $F_4$ and for differences in biomass and carbon turnover for C₃ and C₄ plants, the difference between observed and predicted trends in atmospheric δ¹³CO₂ was only slightly reduced.

## Discussion

We estimated the global fraction of C₄ plants ($F_4$) based on a simple optimality modeling approach driven by climate reanalyses and remote sensing observations to quantify spatiotemporal changes in $F_4$, gross primary production (GPP), and land carbon isotopic discrimination (Δ¹³C) over the period from 1982 to 2016. We also used a simple carbon-cycle box model to determine whether the observed magnitude of the decrease in global atmospheric δ¹³CO₂ could be explained by recent changes in the global abundance and distribution of C₃ and C₄ plants, as suggested by Lavergne2022.

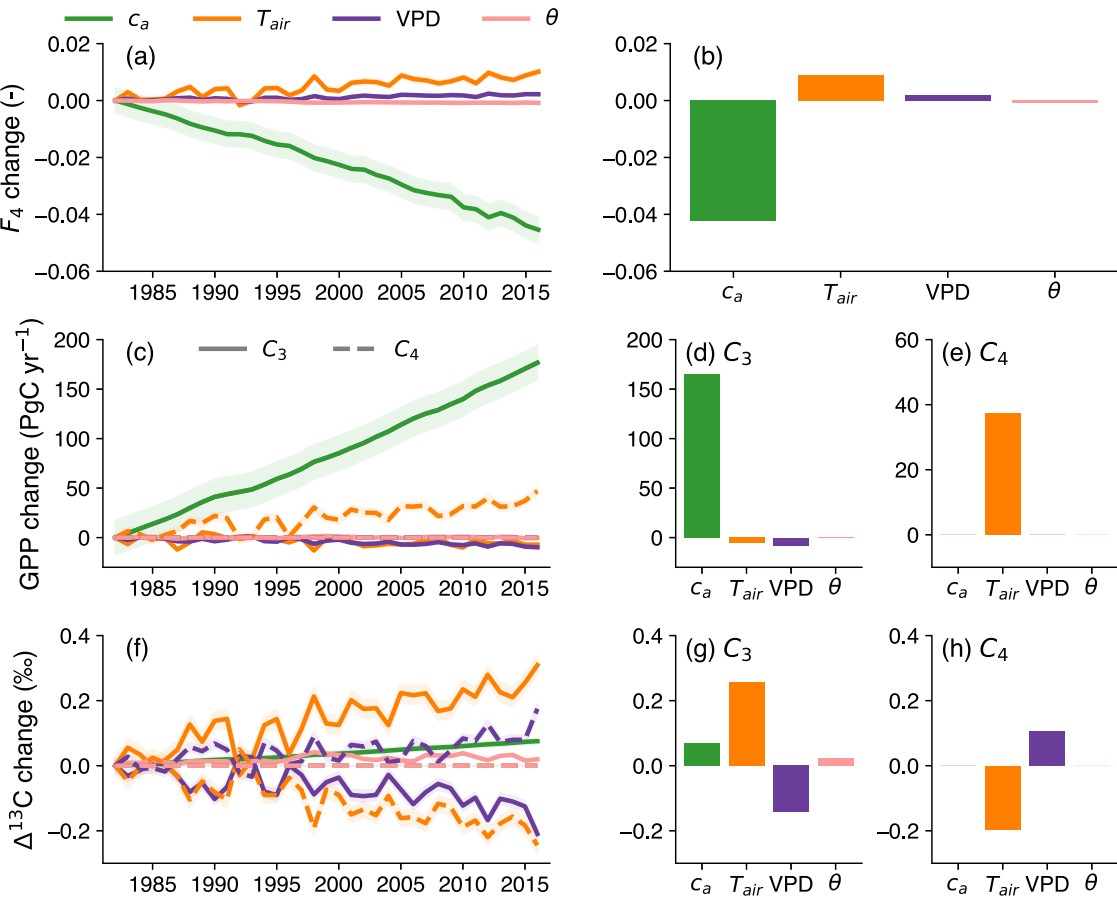

**Fig. 5 | Global averaged change in the impact of environmental drivers on F₄, GPP, and Δ¹³C.** Temporal changes of the impact of atmospheric CO₂ concentrations ($c_a$), daytime air temperature ($T_{air}$), vapor pressure deficit (VPD), and soil moisture (θ) on $F_4$ (**a**), and GPP (**c**) and $\Delta^{13}C$ (**f**) for $C_3$ and $C_4$ plants over 1982–2016.

**b, d, e, g, h** Global average of the individual impacts over the last 5 years (2012–2016). In (**a, c, f**), the light shade represents the 95% confidence interval of the simulated mean.

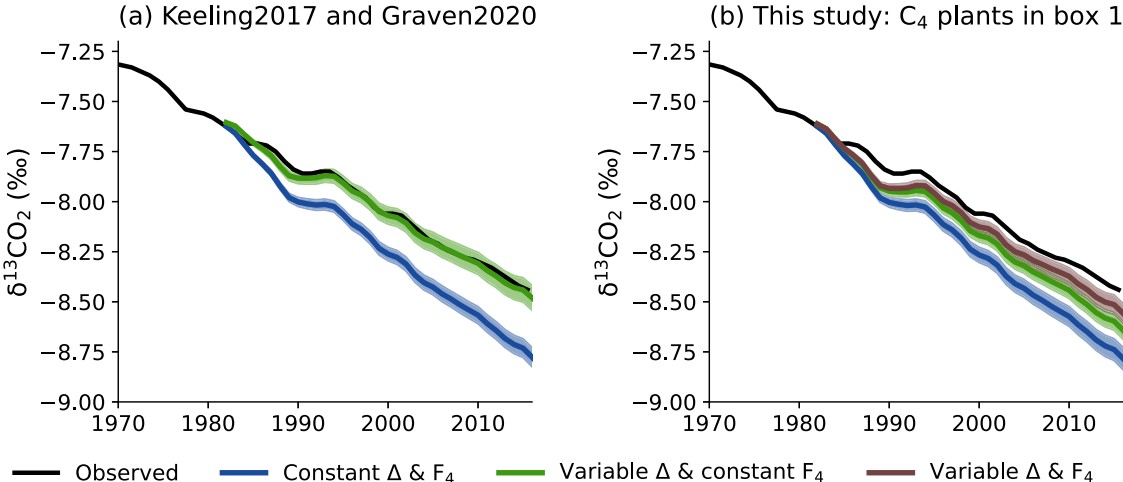

**Fig. 6 | Observed versus predicted δ¹³CO₂ (‰) estimated from a simple carbon cycle model with three biosphere boxes. a** Original model configuration predicting δ¹³CO₂ with simple (blue) and CO₂-driven (green) Δ¹³C as in ref. 3 and ref. 30—denoted Keeling2017 and Graven2020, respectively. **b** Model predictions when box 1 represents only $C_4$ plants, while boxes 2 and 3 are populated only by $C_3$ plants. In blue is the simulation when both Δ¹³C and $F_4$ are constant. In green is the prediction when Δ¹³C is modeled as in Equations S11–S13 with constant $F_4$ and $F_3$. In brown is the simulation when Δ¹³C, $F_4$, and $F_3$ vary.

## Spatiotemporal variations in predicted $F_4$ and differences among approaches

The spatial patterns of $F_4$ predicted by our approach are quite similar to those from Still2009 and Luo2024 over the common period (2001–2016). However, Still2009 tends to overestimate $F_4$ in sub-Saharan and southern Africa and northern Australia compared to our study and that of Luo2024—as also indicated by the higher predicted than observed $\delta^{13}C_{soil}$ values found in Still2009 for these regions (Fig. 1c). $C_4$ plants are important components of African savannas, grasslands, and shrubs[32] and dominate grasslands in the subtropical northeast of Australia[33,34], but they are probably not as overwhelmingly dominant in these regions as the Still2009 map suggests. Both our model and that of Luo2024 capture the relative abundance of $C_3$ and $C_4$ plants in this region. No soil isotope data were available in the dry northeast of Brazil (the Caatinga region) to evaluate the maps; however, the region is known to have a high diversity of both $C_3$ and $C_4$ plants[35,36]. The Still2009 estimate that more than 80% of Caatinga vegetation follows the $C_4$ pathway is thus probably too high, and the co-dominance of $C_3$ and $C_4$ plants suggested by our map and Luo ($F_4$ around 0.5–0.6) seems more realistic.

The Luo2024 map predicts that on average, from 2001 to 2016, $C_4$ plants, including natural grasslands and crops, occupied around 12.5% of the global land surface (excluding deserts and ice-covered lands) and contributed 17.5% of global photosynthesis. These values are slightly lower than those given in the original article (17.5% and 19.5%, respectively) because we considered a larger global land area. Our model estimates are similar to those of Luo2024. We predict that the fraction of $C_4$ plants is 12.7%, contributing 18.5% of global photosynthesis. The value is consistent with previous estimates (18–23%[11,37,38]) and higher than the ensemble mean of dynamic global vegetation models ($14 \pm 13\%$[12]). The predicted decrease in global $F_4$ is greater using our map than using Luo2024, probably because we predicted a decrease in the fraction of $C_4$ in more regions.

## Spatiotemporal changes in GPP and $\Delta^{13}C$ driven by $F_4$ and implications for atmospheric $\delta^{13}CO_2$ variations

The predicted increase in annual total GPP considering both $C_3$ and $C_4$ photosynthesis ($16.5 \pm 1.8$ PgC) from 1982 to 2016 falls within the range of recently published values[21,39]. The $31 \pm 5\%$ increase in GPP over 1900–2010 reconstructed from ice-core records of carbonyl sulfide[40] implies that annual GPP may have increased by around $14 \pm 5\%$ (equivalent to $17.2 \pm 1.5$ PgC) over 1982–2016[21]. Using a light-use efficiency model driven by remote sensing observations[41], suggested an increase in the global GPP of $0.27 \pm 0.02$ Pg C yr$^{-2}$ (or $9.5 \pm 0.7$ PgC yr$^{-1}$, i.e., a 6.1% increase using as reference the mean 1982-1983 values of 115 Pg C yr$^{-1}$) over the period from 1982 to 2015. The spatial distribution of GPP changes highlighted in Fig. S2c is also consistent with that predicted by ref. 21, i.e., an increase in GPP in regions where the $CO_2$ effect on $C_3$ photosynthesis is the largest contributor to the change in GPP[39], such as tropical and European forests. It differs from ref. 41, who suggested a decrease in GPP in large parts of the tropics, but that study ignored the $CO_2$ fertilization effect. Although net carbon uptake in parts of the Amazon rainforest has declined over the past three decades due to increased carbon losses from tree mortality[42], recent estimates of global forest land changes suggest GPP increased in most forests over the past decade[43], in line with our predictions.

When assuming a constant $F_4$, predicted global $\Delta^{13}C$ including both $C_3$ and $C_4$ plants increased by $0.003 \pm 0.001$‰ per ppm increase of $CO_2$ over the 1982–2016 period, consistent with other estimates that ignored changes in the $C_3/C_4$ fraction[2]. However, when the global decrease in $F_4$ over the study period is accounted for, global $\Delta^{13}C$ increased by $0.010 \pm 0.001$‰ ppm$^{-1}$ due to the increase in $\Delta^{13}C$ of $C_3$ plants (and also the slight decrease in $\Delta^{13}C$ of $C_4$ plants). Including crops into this analysis has a minimal effect on $\Delta^{13}C$ and does not significantly change the magnitude of the global $\Delta^{13}C$ trends.

Over their common period (2001–2016), the magnitude of global $\Delta^{13}C$ increase predicted using our $F_4$ map is larger than using Luo2024, despite their relative agreement in predicting global $F_4$ and GPP increase over the recent years. The two maps agree from the late 2000s onwards. The difference could be due to an overestimation of the fraction of $C_3$ plants ($F_3$) across the globe in our map, exacerbating the increasing global $\Delta^{13}C$ trend. We assumed that the sum of $F_3$ and $F_4$ in each land grid point (excluding deserts and snow/glacial lands) is always equal to 1, but this might be an overestimate, as some areas may be covered by other types of land surface.

Using the simple carbon cycle box model, we show that accounting for $\Delta^{13}C$ variations alone—contrary to Keeling2017—cannot explain the observed reduction of the Suess effect in atmospheric $\delta^{13}CO_2$. However, predicted global changes in the abundance and distribution of $C_3$ and $C_4$ plants only slightly reduce the differences between observations and predictions during 2012–2016. The combined effects of increasing global $\Delta^{13}C$ and decreasing fraction of $C_4$ plants reduce the difference from 0.30 to 0.08‰. Thus—contrary to Lavergne2022—our analysis cannot explain the full magnitude of the decrease in atmospheric $\delta^{13}CO_2$, indicating a need to consider other drivers of the isotopic signature of atmospheric $CO_2$. Uncertainties in biosphere processes—especially isotopic fractionation during post-photosynthetic pathways and soil respiration, and carbon residence times in soils—as well as ocean–atmosphere exchanges and fossil fuel emissions, can cause discrepancies between simulated and observed atmospheric $\delta^{13}CO_2$[30,44].

## Limitations of our analysis

The approach to $C_3/C_4$ distribution by Luo2024 is based on a model predicting influences of atmospheric $CO_2$, water stress (soil moisture and vapor pressure deficit), and nitrogen availability on both $C_3$ and $C_4$ photosynthesis[45]. Our model is simpler; it assumes that the impacts of elevated $CO_2$ and water stress are important only for $C_3$ photosynthesis (Supplementary Figs. 11 and 12) and neglects nitrogen availability effects. Nonetheless, our attribution analysis suggests patterns of variability of $F_4$ with elevated $CO_2$ and water stress (Fig. 5) similar to those in Luo2024. Global $F_4$ and GPP values predicted with our $F_4$ model over the recent years are in line with those using the Luo2024 map over their common period (2001–2016). Finally, when comparing measured and predicted $\delta^{13}C_{soil}$, our simple model provides better predictions of $\delta^{13}C_{soil}$ than Luo2024 ($R^2 = 0.58$ versus 0.37), giving us confidence in our predictions.

We considered the photorespiratory effect on $\Delta^{13}C$ trends for $C_3$ plants, but not additional effects (notably the effect of mesophyll conductance), whose importance is debated. Keeling2017 suggested a contribution from the mesophyll conductance effect to the globally increasing $\Delta^{13}C$ trends of $0.006 \pm 0.003$‰ per ppm increase of $CO_2$, which would induce a higher $\Delta^{13}C$ trend in $C_3$ photosynthesis than predicted here: $0.009 \pm 0.003$‰ ppm$^{-1}$ (as opposed to $0.003 \pm 0.001$‰ ppm$^{-1}$) over 1982–2016. On the other hand, Lavergne2022 argued that the mesophyll effect could reduce $\Delta^{13}C$ values and trends by around $-0.001 \pm 0.001$‰ ppm$^{-1}$, so the magnitude of the increase in $\Delta^{13}C$ reported here would then be overestimated. This argument relied on the assumption that the ratio of stomatal to mesophyll conductance is independent of environmental factors, leading to an optimal ratio of the chloroplastic to ambient $CO_2$[27]. This is probably oversimplified; however, the controls of mesophyll conductance are not fully understood.

Finally, when we tested the simple carbon-cycle box model for its sensitivity to changes in the abundance of $C_3$ and $C_4$ plants on atmospheric $\delta^{13}CO_2$, we only incorporated our global $\Delta^{13}C$ estimates for $C_3$ and $C_4$ plants as model inputs, modulated by changes in their relative abundance, carbon use efficiency, and turnover. The standard biosphere carbon model embedded in the box model does not differentiate $C_3$ and $C_4$ photosynthesis and allows the sizes of all three land reservoirs to grow due to $CO_2$ fertilization. Although we removed the $CO_2$ fertilization effect from the first box ($C_4$ photosynthesis), these simplifications of the drivers of $C_3$ and $C_4$ photosynthesis may have biased estimates of net primary production (NPP) and increased uncertainties in modeled carbon dynamics. We also made some assumptions regarding carbon turnover values for the different plant types examined, based on the published parameterization of the carbon cycle box model[30]. For instance, we assume that the carbon turnover for $C_4$

herbaceous (box 1) is about 10 times lower than that of $C_3$ herbaceous (box 2). Although there is ample evidence showing that the turnover time of $C_4$-derived soil carbon is substantially shorter than that of $C_3$-derived soil carbon[46], carbon turnover may still vary across regions. The extent of these variations and potential underlying drivers are still not well constrained.

Regardless of the simplifications in our analysis, however, our results imply that neither of the tested hypotheses[2,3] provide a comprehensive explanation for the observed trend in atmospheric $\delta^{13}CO_2$. Recent shifts in global $\Delta^{13}C$ and the abundance and distribution of $C_3/C_4$ plants can only partially explain the magnitude of the observed decrease in $\delta^{13}CO_2$. Factors other than changes in species abundance and distribution must influence carbon isotope flux exchanges. Further analyses are needed to determine their nature and quantify their contribution to the Suess effect.

## Methods

### Modeling approach based on optimality principles

We used the P model[25–27], a light-use efficiency model based on eco-evolutionary optimality principles, to simulate ecosystem gross primary production (GPP) and carbon isotope discrimination ($\Delta^{13}C$). We also developed a simple scheme based on the P model to predict the share of $C_4$ plants in the total GPP (see Supplementary Note 1 for more details on the model and workflow in Supplementary Fig. 1). We converted the potential share of $C_4$ plants in the total GPP to fraction of $C_4$ plants ($F_{4,pot}$) using the emergent constraint in ref. 12 as the share of $C_4$ plants divided by 1.13.

Since the total fractions of $C_3$ and $C_4$ biomass consist of natural ecosystems (trees, grasses) and crops, we estimated the $C_4$ fraction of natural ecosystems ($F_{4,nat}$) by removing the fraction of human managed areas ($C_3$ and $C_4$ crops and urban areas) from $F_{4,pot}$:

$$F_{4,nat} = F_{4,pot} - \left(F_{3,crops} + F_{4,crops}\right) - F_{urban} \quad (1)$$

We then calculated the total fraction of $C_4$ and $C_3$ plants ($F_{4,tot}$ and $F_{3,tot}$) considering natural ecosystems ($F_{4,nat}$ and $F_{3,nat}$) and crops ($F_{4,crops}$ and $F_{3,crops}$) as:

$$F_{4,tot} = F_{4,nat} + F_{4,crops} \quad (2a)$$

$$F_{3,tot} = 1 - F_{4,tot} = F_{3,nat} + F_{3,crops} \quad (2b)$$

We estimated $C_3$, $C_4$ and total ($C_3 + C_4$) GPP from their respective potential GPP ($GPP_{C3,pot}$ and $GPP_{C4,pot}$) as:

$$GPP_{C3} = GPP_{C3,pot} F_{3,tot} \quad (3a)$$

$$GPP_{C4} = GPP_{C4,pot} F_{4,tot} \quad (3b)$$

$$GPP_{tot} = GPP_{C3} + GPP_{C4} \quad (3c)$$

We aggregated GPP for $C_3$, $C_4$, and all ($C_3 + C_4$) plants from each grid cell, weighted by grid-cell area for each prediction with each $C_4$ map, and compared their respective temporal changes (GPP, in PgC yr$^{-1}$).

We then estimated the land $\Delta^{13}C$ assuming that the biosphere is composed of three terrestrial components ($C_4$ herbaceous, $C_3$ herbaceous and $C_3$ woody) with different carbon turnover times ($\tau$, year) and carbon use efficiencies (CUE, unitless; the ratio of net to gross primary productivity, NPP/GPP):

$$\Delta^{13}C = \frac{f_{4,herb}\, \Delta_4\, \tau_{4,herb} + f_{3,herb}\, \Delta_3\, \tau_{3,herb} + f_{3,woody}\, \Delta_3\, \tau_{3,woody}}{\tau_{4,herb} + \tau_{3,herb} + \tau_{3,woody}} \quad (4)$$

with $\tau_{4,herb}$, $\tau_{3,herb}$, $\tau_{3,woody}$ the carbon turnover times and $f_{4,herb}$, $f_{3,herb}$ and $f_{3,woody}$ the fractions adjusted by CUE ($CUE_{4,herb}$, $CUE_{3,herb}$ and $CUE_{4,woody}$) for $C_4$ herbaceous, $C_3$ herbaceous and $C_3$ woody plants

respectively.

$$f_{4,herb} = F_{4,tot,t0} - \left(F_{4,tot,t0} - F_{4,tot}\right) CUE_{4,herb} \quad (4a)$$

$$f_{3,herb} = F_{3,tot,t0} - \left(F_{3,tot,t0} - F_{3,tot}\right) CUE_{3,herb} \quad (4b)$$

$$f_{3,woody} = F_{3,tot,t0} - \left(F_{3,tot,t0} - F_{3,tot}\right) CUE_{3,woody} \quad (4c)$$

$F_{4,tot,t0}$ and $F_{3,tot,t0}$ are the fractions of $C_4$ and $C_3$ plants, respectively, at time $t0$ - the first year of the records (1982).

$\Delta_4$ and $\Delta_3$ are the carbon isotope discrimination of $C_4$ and $C_3$ plants, respectively (see Supplementary Note 1), calculated for each month ($mth$) and weighted by each month's $C_3$ and $C_4$ GPP to obtain annual averages of $\Delta_3$ and $\Delta_4$:

$$\Delta_{x,yr} = \sum_t \frac{\Delta_{x,mth} \times GPP_{x,mth}}{\sum_t GPP_{x,mth}} \quad (5)$$

with $x$ being the plant type ($C_3$ or $C_4$) selected.

On average, biomes dominated by $C_4$ plants have both higher carbon turnover rates and lower biomass than those dominated by $C_3$ plants[46–48], so their relative contribution to land $\Delta^{13}C$ is expected to be lower than that of $C_3$ plants. $\tau$ values estimated as the ratio of carbon stocks in vegetation and soils to NPP from TRENDYv12 model outputs[49] vary strongly among models. For instance, global average $\tau$ (median ± standard deviation) are 8.7 ± 11.2, 22.4 ± 34.2 and 15.3 ± 27.5 year for $C_4$ grasses, 33.8 ± 33.3, 40.5 ± 81.1 and 534.3 ± 217.7 year for $C_3$ grasses, and 561.6 ± 173.7, 616.0 ± 224.3, and 641.3 ± 181.2 year for $C_3$ woody, respectively for CABLE-POP, CLASSIC, and ORCHIDEE models. Given the large uncertainties in these estimates and to ensure consistency with our whole approach, we use the same range of $\tau$ values as in a carbon cycle box model parameterization when the $CO_2$ fertilization effect is on for $C_3$ plants[30] ($\tau_{4,herb} = 2.4 \pm 0.3$ year, $\tau_{3,herb} = 24.4 \pm 6.5$ year, and $\tau_{3,woody} = 299.4 \pm 144.5$ year; see also Supplementary Table 1). These values are at the lower end of the estimates from the three TRENDYv12 models but are consistent with the relative differences in $\tau$ between $C_4$ herbaceous, $C_3$ herbaceous and $C_3$ woody plants (very fast, intermediate, and slow, respectively).

Non-forest biomes, including grasslands, shrublands, and crops, tend to have higher CUE than forests (0.46 ± 0.11 versus 0.41 ± 0.11), except for $C_4$ grass-dominated savanna ecosystems, which show a lower CUE (0.32 ± 0.12)[50]. However, the contribution of $C_3$ and $C_4$ plants in grass-lands, shrublands, and crops, and their relative CUEs remain unclear. Here, we use the average CUE values reported in reference 46, assuming that $C_4$ biomass is a mixture of grasslands, savanna ecosystems, and crops (0.41 ± 0.10), that $C_3$ herbaceous biomass is a mixture of grasslands and crops only (0.46 ± 0.10), and that $C_3$ woody biomass includes both forests and shrubland biomes (0.40 ± 0.10)[46]. We acknowledge the limitations of our approach, especially given that $\tau$ and CUE may vary not only across plant functional types but also across environmental conditions, in particular temperatures[50–53].

### Optimality model configuration and simulation

We ran the P-model on a 0.5° resolution grid at a monthly timestep over the period 1982–2016, forced by monthly mean values of daytime air temperature ($T_{air}$, °C), daytime vapor pressure deficit (VPD, Pa), incident photosynthetic photon flux density (PPFD, mol m$^{-2}$ month$^{-1}$), the fraction of incident PPFD absorbed by foliage ($f$APAR, dimensionless), the atmospheric partial pressure of $CO_2$ ($c_a$, Pa), root-zone volumetric soil moisture ($\theta$, m$^3$ m$^{-3}$) and elevation ($z$, m). Monthly mean $T_{daytime}$ and VPD were calculated from minimum and maximum temperature and actual vapor pressure from the Climatic Research Unit (CRU) gridded time-series (CRU TS4.03) dataset[54] to consider only the part of the day when photosynthesis occurs[55]. Monthly shortwave downwelling radiation ($SWdown$) was obtained from WATCH-Forcing-Data-ERA-Interim (WFDEI) data[56] and

used to calculate monthly PPFD. Supplementary Table 2 summarizes the data information and sources used in this study.

Monthly $f$APAR data were derived from the Advanced Very High Resolution Radiometer (AVHRR) Global Inventory Modeling and Mapping Studies (GIMMS) $f$APAR 3 g product[57] gridded at 0.5° resolution. Since monthly $c_a$ varies spatially, we used the annual $c_a$ data (μmol mol$^{-1}$) derived from ref. 58 and converted them into Pa using elevations derived from WATCH-WFDEI. Monthly θ (m$^3$ m$^{-3}$) over a 1 m soil depth was calculated using a modified version of the SPLASH model[59] driven by daily precipitation, $T_{air}$ and $SWdown$ from the WATCH-WFDEI dataset over 1979–2018. The model θ outputs were derived from ref. 60.

Annual data on percentage tree cover, used to estimate the fraction of $C_4$ plants (see Text S1), were derived from the NASA Making Earth System Data Records for Use in Research Environments (MEaSURES) Vegetation Continuous Fields (VCF) 5KYR v001[61] for the 1982–2016 period. The years 1994 and 2000 are missing from this dataset. We interpolated tree cover for these years by averaging values from the previous and subsequent years (1993 and 1995, and 1999 and 2001, respectively).

We used the urban areas and $C_3$ and $C_4$ crop distribution estimates from the LUHv2-2019 dataset (https://daac.ornl.gov/VEGETATION/guides/LUH2_GCB2019.html). Data for the remote-sensing products given at 0.05° resolution were aggregated to 0.5° resolution using the mean of all the 0.05° grid cells within the 0.5° grid cell. Climate and remote sensing data were filtered with the MODIS Land Processes Distributed Active Archive Center (LP DAAC; https://www.earthdata.nasa.gov/data/catalog/lpcloud-mcd12q1-061) snowandice and barren_sparsely_vegetated maps to retain only vegetated regions free of snow and ice.

### Optimality model evaluation and comparison

We evaluated the simulations of $\Delta^{13}C$ for $C_3$ and $C_4$ plants using a compilation of leaf stable carbon isotope data[28] that includes 3601 measurements for $C_3$ plants and 531 measurements for $C_4$ plants (see Supplementary Fig. 8a for site locations). Measured leaf $\delta^{13}C$ were converted into leaf $\Delta^{13}C$ using atmospheric $\delta^{13}CO_2$ from ref. 62. We also evaluated the model predictions of the fraction of $C_4$ plants ($F_4$) over the 1982–2016 period using a new compilation of 2156 soil $\delta^{13}C$ measurements derived from published sources specifically compiled for this study[29] (see Supplementary Table 3 and Supplementary Fig. 8b for data information and locations). Predicted $\Delta^{13}C$ values for $C_3$ and $C_4$ plants were converted to $\delta^{13}C$ using atmospheric $\delta^{13}CO_2$ from ref. 62. Since measured soil $\delta^{13}C$ is an average of the stable carbon isotopic signature of soil organic matter accumulated over several years, we estimated soil $\delta^{13}C$ from the model as the average of predicted $\delta^{13}C$ for both $C_3$ and $C_4$ plants, weighted by their relative yearly fraction, over the entire period 1982–2016 to enable comparisons with the observations. We assumed that any additional isotopic fractionation within the soil is negligible over the study timeframe.

We compared the predictive skills of our approach with those from two independent, global $C_4$ distribution maps. We used the map from ref. 20 available at https://daac.ornl.gov/cgi-bin/dsviewer.pl?ds_id=932 and the map recently developed by ref. 12 for the period 2001–2019 (https://zenodo.org/records/10516423)—referred to as Still2009 and Luo2024, respectively. Since these maps covered different regions of the world, we homogenized them to make them comparable. When grid points were set to "NA" but were on land (excluding deserts and snow/glacial lands), we assumed $F_4 = 0$. As a result, the estimated $F_4$ values derived from the maps may be slightly different from those reported in the respective publications. We predicted soil isotopic composition using each of the maps and compare them with the soil isotopic network. We also compared the mean and trends in total $F_4$, GPP, and $\Delta^{13}C$ derived from each of the maps.

We conducted additional simulations to quantify the contributions of $c_a$, $T_{air}$, VPD and θ to $F_4$, GPP and $\Delta^{13}C$ changes. To do so, we ran the model for four different scenarios in which we used the input of $c_a$, $T_{air}$, VPD and θ of the first year (1982), respectively, to simulate GPP over the whole period. We then used predicted GPP for the different scenarios to estimate $F_4$ for

each year. We calculated the difference between the original model simulations and those derived from each scenario to determine the relative contributions of $c_a$, $T_{air}$, VPD and θ to $F_4$, GPP, and $\Delta^{13}C$ changes.

### Simple carbon cycle model to estimate atmospheric $\delta^{13}CO_2$

To account for differences in land $\Delta^{13}C$, biomass and carbon turnover between $C_3$ and $C_4$ plants, we estimated global average atmospheric $\delta^{13}CO_2$ (‰) over 1982–2016 using the simple carbon cycle model from ref. 30, also used in ref. 3. The model simulates carbon cycling in atmospheric, oceanic, and biospheric reservoirs, and includes one atmospheric box, three biospheric boxes with different biomass and carbon turnover times, and a one-dimensional box diffusion ocean model with 43 ocean boxes. We conducted our historical simulations using the same initial model configuration and calibrated parameter ranges as in ref. 30. A few small changes were made to the model code: defining different $\Delta^{13}C$ values for $C_3$ and $C_4$ plants that consider temporal changes in the fraction of $C_3$ and $C_4$ plants, and different carbon turnover (τ) and use efficiency (CUE) for the three biospheric boxes (see also Supplementary Note 1 for more information). We first ran the model in its standard mode, initially with a constant $\Delta^{13}C$ (equal to 18‰) and then with a variable $\Delta^{13}C$ based on $CO_2$ changes as in ref. 3. We then tested the model using our global average annual model outputs for $\Delta^{13}C$ and $F_4$ by allowing box 1, with low biomass and rapid τ, to represent only $C_4$ herbaceous, while boxes 2 and 3, with intermediate and high biomass and intermediate and slow τ, represent $C_3$ herbaceous and woody, respectively. Since $CO_2$ fertilization only impacts $C_3$ photosynthesis, we assume it to be null for $C_4$ photosynthesis (box 1).

### Data availability

The data that support the findings of this study are publicly available. The CRU TS4.03 datasets are available from East Anglia University (UK) at https://crudata.uea.ac.uk/cru/data/hrg/. The WATCH-WFDEI dataset is available from the International Institute for Applied Systems Analysis (Austria) via the WATCH FTP server at ftp://rfdata:forceDATA@ftp.iiasa.ac.at. The annual percentage treecover from MEaSURES VCF5KYR v001 is available at https://doi.org/10.5067/MEaSUREs/VCF/VCF5KYR.001. Urban areas and $C_3$ and $C_4$ crop distribution from LUHv2-2019 data are available at https://daac.ornl.gov/VEGETATION/guides/LUH2_GCB2019.html. The snowandice and barren_sparsely_vegetated landcover maps from MODIS LP DAAC are available at https://www.earthdata.nasa.gov/data/catalog/lpcloud-mcd12q1-061. The map of fraction of $C_4$ plants from ref. 20 is available at https://doi.org/10.3334/ORNLDAAC/932. The global $C_4$ distribution map developed by ref. 12 is available at https://zenodo.org/records/10516423. The AVHRR GIMMS $f$APAR data were made available by R. Myneni (data request contact: rmyneni@bu.edu). The concentrations and isotopic compositions of atmospheric $CO_2$ are available in the Supplementary Material of refs. 30,58,62. The soil carbon isotopic data were extracted from ref. 29 and is available at https://doi.org/10.5281/zenodo.6556096. The leaf carbon isotopic data are derived from ref. 28, the leaf gas-exchange data for $C_4$ plants from ref. 63 and the share of $C_4$ plants in ecosystem GPP from refs. 64,65, all available in the original papers. The processed climate inputs and global model outputs produced in this article are available at https://zenodo.org/records/17726762.

### Code availability

The original P-model was incorporated into Python (https://pyrealm.readthedocs.io/en/latest/), and the code modifications and additions presented in this paper are available at https://pyrealm.readthedocs.io/en/latest/users/pmodel/c3c4model.html. The Python code to run the new $C_3$/$C_4$ competition model, analyse simulations and create the figures is available at https://github.com/Alielav/Comms-Earth-Lavergne-et-al.-2025. The original MATLAB code to run the simple carbon cycle box model and predict atmospheric $\delta^{13}CO_2$ is available from Heather Graven at https://github.com/heathergraven/simplemodel2020. The Python version that includes the code edits made for the purpose of this article is also available at https://github.com/Alielav/Comms-Earth-Lavergne-et-al.-2025.

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

## Acknowledgements

This research is a contribution to the LEMONTREE (Land Ecosystem Models based On New Theory, obseRvations and ExperimEnts) project, funded by Schmidt Futures LLC (G-21-61881) (A.L., S.P.H., I.C.P.). A.L., S.P.H., and K.A. acknowledge support from the ERC-funded project GC2.0 (Global Change 2.0: Unlocking the past for a clearer future, grant number 694481). I.C.P. and N.D. acknowledge support from the ERC-funded project REALM (Re-inventing Ecosystem And Land-surface Models, grant number 787203). We thank R. Myneni and Z. Zhu for providing the AVHRR GIMMS fAPAR dataset, and the many researchers who have made their stable carbon isotope and plant C4 fraction data publicly available. We also thank David Orme for incorporating the new code into the official Python version of the P-model, Heather Graven for suggesting the use of the simple carbon cycle model and providing guidance for running it, and Joseph Ovwemuvwose for helpful discussions on crop inclusions in the analyses.

## Author contributions

I.C.P. and S.P.H. proposed an initial idea and general approach with further development from A.L. and K.A. took the first steps in developing the $C_3/C_4$ model. D.N. provided the compiled soil $\delta^{13}C$ data set with input from A.L. A.L. completed the work, including selecting and expanding the soil $\delta^{13}C$ data set, and designing and executing model evaluations and comparisons, and writing the manuscript draft. A.L. also incorporated the $C_3/C_4$ model code into the Python version of the P model (pyrealm) and translated the original MATLAB code of the simple carbon cycle model into Python. All authors contributed to the interpretation of the results and the final draft.

## Competing interests

The authors declare the following competing interests: Aliénor Lavergne was a Senior Editor at Communications Earth & Environment until 14 August 2025 and is now a Senior Editor at Nature Geoscience, but was not involved in the editorial review of, nor the decision to publish this article at any point.
