## [Transparent Peer Review file · Communications Earth & Environment]

Minimal impact of the recent decline in C₄ vegetation abundance on atmospheric carbon isotopic composition

Corresponding Author: Dr Aliénor Lavergne

Version 0:

Decision Letter:

Dear Dr Lavergne,

Your manuscript titled "Recent C₄ vegetation decline is imprinted in atmospheric carbon isotopes" has now been seen by 2 reviewers, and we include their comments at the end of this message. They find your work of interest, but some important points are raised. We are interested in the possibility of publishing your study in Communications Earth & Environment, but would like to consider your responses to these concerns and assess a revised manuscript before we make a final decision on publication.

We therefore invite you to revise and resubmit your manuscript, along with a point-by-point response that takes into account the points raised. In particular, we require that you:

- comprehensively validate and evaluate model performance,
- estimate model sensitivity to relevant key environmental parameters,
- appropriately tone down, caveat or remove unsupported claims to reflect limitations and uncertainties in your analysis.

Please highlight all changes in the manuscript text file.

Please use the following link to submit your revised manuscript, point-by-point response to the referees' comments (which should be in a separate document to any cover letter), a tracked-changes version of the manuscript (as a PDF file) and the completed checklist:

Link Redacted

We hope to receive your revised paper within six weeks; please let us know if you aren't able to submit it within this time so that we can discuss how best to proceed. If we don't hear from you, and the revision process takes significantly longer, we may close your file. In this event, we will still be happy to reconsider your paper at a later date, as long as nothing similar has been accepted for publication at Communications Earth & Environment or published elsewhere in the meantime.

Please do not hesitate to contact us if you have any questions or would like to discuss these revisions further. We look forward to seeing the revised manuscript and thank you for the opportunity to review your work.

Best regards,

Clare Davis, PhD
Senior Editor
Communications Earth & Environment

EDITORIAL POLICIES AND FORMATTING

Editorial Policy: [Policy requirements](https://www.nature.com/documents/nr-editorial-policy-checklist.pdf) (Download the link to your computer as a PDF.)

- Behavioural and social science
- Ecological, evolutionary & environmental sciences
- Life sciences

<https://www.nature.com/documents/nr-reporting-summary.zip>

Furthermore, please align your manuscript with our format requirements, which are summarized on the following checklist: [Communications Earth & Environment formatting checklist](https://www.nature.com/documents/commsj-phys-style-formatting-checklist-article.pdf)

and also in our style and formatting guide [Communications Earth & Environment formatting guide](https://www.nature.com/documents/commsj-phys-style-formatting-guide-accept.pdf).

*** DATA: Communications Earth & Environment endorses the principles of the Enabling FAIR data project (<http://www.copdess.org/enabling-fair-data-project/>). We ask authors to make the data that support their conclusions available in permanent, publically accessible data repositories. (Please contact the editor if you are unable to make your data available).

All Communications Earth & Environment manuscripts must include a section titled "Data Availability" at the end of the Methods section or main text (if no Methods). More information on this policy, is available at <http://www.nature.com/authors/policies/data/data-availability-statements-data-citations.pdf>.

If a community resource is unavailable, data can be submitted to generalist repositories such as [figshare](https://figshare.com/) or [Dryad Digital Repository](http://datadryad.org/). Please provide a unique identifier for the data (for example a DOI or a permanent URL) in the data availability statement, if possible. If the repository does not provide identifiers, we encourage authors to supply the search terms that will return the data. For data that have been obtained from publically available sources, please provide a URL and the specific data product name in the data availability statement. Data with a DOI should be further cited in the methods reference section.

REVIEWER COMMENTS:

Reviewer #1 (Remarks to the Author):

This study, which builds upon Lavergne et al (2022), presents results of a model (P model) that estimates the distribution of C3 and C4 vegetation, allowing assessments of changes in the C3/C4 distribution and impacts on GPP and isotopic discrimination. The model predicts that rising CO₂ has driven a decrease in C₄, which drives an upwards trend in global discrimination and modifies the trajectory in GPP. The discrimination increase agrees well with an independent estimate of

the trend in global discrimination based on the atmospheric $\delta^{13}\text{C}$ trend from Keeling et al. (2017).

The paper is overall well written, makes some interesting points, and is useful in presenting detailed results from the P model. The main issue is that it overlooks some important caveats.

One caveat involves the comparison with the Keeling et al (2017). A change in discrimination can impact the atmospheric $\delta^{13}\text{C}$ trend only to the extent that it isotopically alters carbon stored in vegetation and soils. The land biospheric impact on the global $\delta^{13}\text{C}$ budget thus involves a convolution of discrimination AND carbon turnover. Photosynthesis which feeds a short-lived pool has less leverage than photosynthesis which fuels a long-lived pool. In this context, it's highly relevant that the turnover of C4 carbon is generally more rapid than turnover of C3 carbon, as C4 is rare in trees. The model used in Keeling et al did not address differential turnover of C3 versus C4, but the issue was addressed briefly in discussion, where the impact of C4 was discounted. In any case, the discrimination trend estimated in Keeling et al (2017) should not be naively interpreted as a GPP-weighted average of C3 and C4 discrimination. Rather, it must be weighted towards C3, possibly quite heavily. Clearly more work is needed on this topic, e.g. one needs a model that includes carbon turnover in a range of pools divided by C3 and C4. For now, the authors need to at least point out this important caveat.

An accounting of differences in the turnover of C3 versus C4 carbon is also needed to support the comparisons that are presented between the P model and measurements of $\delta^{13}\text{C}$ in soil organic matter (Figure 1). How was this addressed? If this was not addressed, some caveats are needed. See Wynn and Bird (2007, <https://doi.org/10.1111/j.1365-2486.2007.01435.x>).

An important caveat on the use of tree rings to infer long-term trends in discrimination is that these studies (to my knowledge) have not yet addressed a range of complications that parallel those that arise using tree rings to infer long-term trends in growth (Brienen et al, doi: 10.1111/gcb.13605).

Some further discussion is merited on mesophyll impacts on discrimination. The current version of the p-model neglects mesophyll impacts on discrimination, which the authors justify based on the Lavergne (2022) et al study suggesting that mesophyll impacts have little impact on the discrimination in response to rising CO_2 . As far as I can tell, the Lavergne (2022) result is tied to a built-in assumption that mesophyll conductance tends to scale in proportion to stomatal conductance in response to rising CO_2 . This is an interesting hypothesis, but is this really a settled issue? What are the bounds in its validity? It would seem appropriate to add the caveat that the uncertainty arising from this assumption has not been addressed, and suggest further work to reduce these uncertainties.

Maybe the most efficient way to address these caveats is to add a section at the end on needed follow-up studies.

Minor points:

49. Whether there is a global trend or not needs to be decided by observations not models. The jury is still out, I think.

Figure 1. It might help to put "observed" within Figure 1b, paralleling 1c and 1d.

Figure 2. On 2b and 2c, add the word "from" as in "from Still2009". For 2a, also add "from..." specifying the model name.

99: Higher than what?

Figure 3. Specify model and time period in all three panels.

115: The basis of GPP calculation is not entirely clear from the wording. Per unit surface area?. Maybe this needs a more precise term than "GPP". The wording needs to allow that the reader may not yet have read the Method section.

Eq (1) and Eq. (2a) appear to use inconsistent notation $F_{4,\text{nat}}$ versus F_4 , natural.

301. spell out Ref 43 to complete the sentence.

Reviewer #2 (Remarks to the Author):

The authors use the P light use efficiency model combined with a C3/C4 competition model to demonstrate that the recent increase (2001-2016) in natural C3 landcover is likely responsible for increased $\delta^{13}\text{C}$ photosynthesis globally and the reason for the decreasing trend of $\delta^{13}\text{C}$ atmosphere overall. As validation for their model approach, the reviewers compare their C3/C4 model and simulated GPP against Still and Luo approaches, claiming the Still approach significantly overestimates C4 landcover overall. They use site level $\delta^{13}\text{C}$ leaf and $\delta^{13}\text{C}$ soil estimates as validation to their model approach.

Although this reviewer found the author's C3/C4 landcover mechanism a plausible explanation for the decreased trend in $\delta^{13}\text{C}$ atmosphere there were two significant concerns. First the validation of the P model and C3/C4 competition model was very limited. It would have been more compelling if the P model was validated against GPP measurements at flux tower sites or against global reanalysis products like FLUXCOM (Jung et al.). The author's do supply Figure 1a as a type of validation for the photosynthetic discrimination model, however, it does seem like the model underestimates the magnitude of C3 discrimination when compared to the leaf observations, and thus might be compensating for this bias with increased C3 landcover. Second, the authors do not discuss to what extent trends in VPD (which influence discrimination) may have impacted the atmospheric signature. To what extent did your meteorological forcing product show trends in VPD and soil moisture and how did this influence the discrimination in the P model? The authors provide a brief discussion of model GPP sensitivity to environmental variables in the supplement, but the sensitivity of the model photosynthetic discrimination is lacking. Detailed comments below:

Abstract:

"We conclude that the magnitude of the decrease in global atmospheric $\delta^{13}\text{C}_{\text{CO}_2}$ can be partly explained by global changes in the distribution of C3/C4 plants."

This implies that C4 natural grasslands are strongly decreasing. Does it make sense that the CO₂ concentration increase leads to this given increased aridity which favors C4 plants?

Introduction:

"A more recent work has incorporated C3/C4 adaptation and acclimation to recent environmental changes based on an optimality model and observations. The derived map suggests that the global fraction of C4 plants has decreased over 2001-2019 period due to a decrease in C4 natural grasses with elevated CO₂, even though the abundance of C4 crops has increased."

I feel like there needs to be a bit more explanation of why C3 plants discriminate more against $\delta^{13}\text{C}$ as compared to C4 plants. More background. No explanation of alternative hypotheses to C3/C4 land surface transition. See Raczka et al., 2017 JGR-Biogeosciences.

Also, I feel there is a lack of background given to post-photosynthetic fractionation processes in general, that could influence $\delta^{13}\text{C}$ soil (See Bruggemann et al., 2011)

Methods:

Authors evaluated the light-use efficiency (P model) against leaf isotope data for C3 and C4 plants, but what about GPP and NPP predictions based on flux tower data, NEON site data or global GPP products like FLUXCOM? Has the carbon model within the p model been evaluated/validated at all? You are looking at delphoto, and not delland atmosphere exchange?

Results:

Figure 3: I don't understand how F4 trends can be decreasing everywhere, all across the globe. Especially when the authors state F4 is increasing over agriculture areas. This assumes the effect of CO₂ fertilization is superseding any drying/warming trends everywhere. Seems highly unlikely.

Discussion:

The only validation has been with sporadic site level leaf $\delta^{13}\text{C}$ for C3 and C4 species (Fig1). When the C3 and C4 species are considered individually the skill is quite modest, but it looks like they are comparing coarse grid cell (0.5x0.5) against site level data. The spatial mismatch is significant and the meteorological forcing mismatch impedes a fair comparison.

P Optimality Model: Figure S1 attempts to explain how it is constrained. The Paruelo based logistic regression looks poorly fit to the observations. There is no mention how the uncertainty between Adv4 and Sh4 contribute to uncertainty in the prediction.

The authors claim that increases in CO₂ are allowing for C3 species to outcompete C4 species, however, the authors do not discuss the impact of VPD and soil moisture have on C3/C4 competition. Warmer, drier conditions tends to favor C4 species. Furthermore the authors show their P optimality model is highly sensitive to VPD (much more so than CO₂) in terms of GPP – these implies stomatal conductance is strongly reduced which should increase discrimination in C3 species. Could simply a stable C3/C4 spatial map, combined with increased discrimination of C3 species account for trends in atmospheric $\delta^{13}\text{C}$? (No, Lavergne tried this in a previous manuscript, but could this be because of underestimated C3 discrimination?)

The fact that their C3/C4 competition model is showing increased C3 coverage with time *everywhere* across the globe, with a model that is way more sensitive to VPD, than CO₂ (Figure S11) , indicates to me that the met forcing product they used is showing stable or increased moisture across this period. Did they check this for trends? They need more investigation into their met forcing product. I think more explanation of the C3 fractionation results and sensitivity to VPD s

necessary. To show the sensitivity of C4 plants to c_i/c_a (Figure S8), but no mention of sensitivity of C3 discrimination to environmental drivers seems odd.

Why wouldn't you show sensitivity of C3 and C4 to $\delta^{13}C$ discrimination based on CO₂, VPD, and soil moisture as well (Figure S11)? The authors show sensitivity to GPP, but don't actually validate the model at all against GPP site level or regional level (FLUXCOM) products. They show limited $\delta^{13}C$ leaf and $\delta^{13}C$ soil, but seem to focus on the $\delta^{13}C_{soil}$ results which are less related to the photosynthetic fractionation than leaf $\delta^{13}C$. The $\delta^{13}C$ of soil is a function of $\delta^{13}C$ photosynthesis and $\delta^{13}C$ respiration and other post fractionation processes, which the P model does not include (Bruggemann et al., 2011)

Line. 161: "However, Still2009 tends to overestimate F4 in sub-Saharan and southern Africa and northern Australia compared to our map and that of Luo2024 - as also indicated by the higher predicted than observed $\delta^{13}C_{soil}$ values found in Still2009 for these regions (Figure 1c)."

But isn't $\delta^{13}C$ leaf a better indicator of fractionation from photosynthesis as compared to $\delta^{13}C$ of soil, which could be influenced by post-photosynthetic fractionation processes? Could you not show Figure 1A compared to Still and Luo?

Line 197: "Nevertheless, results from our sensitivity analysis suggest similar patterns of variability of F4 with elevated CO₂ and water stress as with Luo2024 map, i.e., small negative effect of elevated CO₂ and positive effect of high VPD and low soil moisture on F4 (Text S2 and Figure S11)"

I don't understand this. Figure S11 shows dominant impact of VPD on GPP and likely discrimination, which would seem to allow C4 species to outcompete C3. The authors suggest the opposite in the intro, that trends in CO₂ are overwhelming the F4 trend.

Line 222: Have you demonstrated the simulated $\delta^{13}C$ for C3 species is accurate? Has it been validated at the site level or any type of observation that matches the spatial scale of your simulation?

Line 247: "Our study highlights the importance of considering recent C3 and C4 land cover changes with elevated atmospheric CO₂ and increasing water stress in the terrestrial carbon budget and pave the way for an improved evaluation of the mechanisms at play."

These authors did not consider/discuss the impact of water stress on the trends in GPP and $\delta^{13}C$. This was surprising considering the high sensitivity of the P model to VPD, and suggests drought tolerant species C4, could maintain or perhaps expand their extend in a warmer/drier world. The authors need to discuss the pattern of VPD trends by region and over time, and how that impacted their results.

Communications Earth & Environment is committed to improving transparency in authorship. As part of our efforts in this direction, we are now requesting that all authors identified as 'corresponding author' create and link their Open Researcher and Contributor Identifier (ORCID) with their account on the Manuscript Tracking System prior to acceptance. ORCID helps the scientific community achieve unambiguous attribution of all scholarly contributions. You can create and link your ORCID from the home page of the Manuscript Tracking System by clicking on 'Modify my Springer Nature account' and following the instructions in the link below. Please also inform all co-authors that they can add their ORCID to their accounts and that they must do so prior to acceptance.

Version 1:

Decision Letter:

Dear Dr Lavergne,

Your revised manuscript titled "Recent C₄ vegetation decline impacts global carbon isotopic discrimination" has been seen by the original two reviewers, whose comments are appended below. You will see that they find your work of some potential interest. However, they have raised substantial concerns that must be addressed. In light of these comments, we cannot accept the manuscript for publication, but would be interested in considering a revised version that fully addresses these serious concerns.

We hope you will find the reviewers' comments useful as you decide how to proceed. Specifically, for publication in Communications Earth & Environment to be appropriate, a further revised manuscript would need to provide compelling support for their conclusion that globally the fraction of C₄ plants has substantially decreased between 1982 and 2016, whereas the fraction of C₄ crops has increased, addressing in full the reviewers' queries about the validity of the model and inconsistencies in the description of the methodology.

Should additional work allow you to meet this editorial threshold and all the reviewers' criticisms, we would be happy to look at a substantially revised manuscript. If you choose to take up this option, please either highlight all changes in the manuscript text file, or provide a list of the changes to the manuscript with your responses to the reviewers.

When resubmitting, please provide a point-by-point response to the reviewers' comments. Please submit your responses as a separate file, distinct from your cover letter where you can add responses to the Editors' comments that you do not want to be made available to the reviewers. Word files are preferred.

Important: The response to reviewers should not include any figures, tables or graphs. If you wish to respond to the reviewer reports with additional data in one of these formats, please add them to the main article or Supplementary Information, and refer to them in the rebuttal.

If the revision process takes significantly longer than three months, we will be happy to reconsider your paper at a later date, as long as nothing similar has been accepted for publication at Communications Earth & Environment or published elsewhere in the meantime.

Please use the following link to submit your revised manuscript, point-by-point response to the reviewers' comments with a list of your changes to the manuscript text (which should be in a separate document to any cover letter), a tracked-changes version of the manuscript (as a PDF file) and any completed checklist:

Link Redacted

Please do not hesitate to contact us if you have any questions or would like to discuss the required revisions further. Thank you for the opportunity to review your work.

Best regards,

Heike Langenberg, PhD
Chief Editor
Communications Earth & Environment

On X(Twitter): @CommsEarth

EDITORIAL POLICIES AND FORMAT

If you decide to resubmit your paper, please ensure that your manuscript complies with our editorial policies and complete and upload the checklist below as a Related Manuscript file type with the revised article:

Editorial Policy Policy requirements
(Download the link to your computer as a PDF.)

- Behavioural and social science
- Ecological, evolutionary & environmental sciences
- Life sciences

<https://www.nature.com/documents/nr-reporting-summary.zip>

For your information, you can find some guidance regarding format requirements summarized on the following checklist:

(<https://www.nature.com/documents/commsj-phys-style-formatting-checklist-article.pdf>) and formatting guide

(<https://www.nature.com/documents/commsj-phys-style-formatting-guide-accept.pdf>).

REVIEWER COMMENTS:

Reviewer #1 (Remarks to the Author):

I laud the work done to address the issue of differences in turnover time between C3 and C4 carbon.

I note, however, that Eq. 6 looks fishy on mathematical grounds. A reality check is that Δ_{atm} remains bounded between the pure C4 and C3 limits as τ grades from zero to infinity. But according to Eq. (6), if τ is infinite, then Δ is infinite. And $\tau=0$ also misses the pure C3 limit.

I also note that timescale of the factors driving changes in discrimination (e.g. the timescale of the CO₂ rise, or climate change) will dictate which carbon pools (long-lived versus short-lived) are most important. Even a very simple treatment of differential C₄/C₃ turnover in the context of changing discrimination therefore needs to include information on how fast discrimination is changing. I would be tempted to start by trying to understand the behavior of a one-box carbon model with exponentially changing discrimination ($\Delta = a + b e^{(t/T)}$) with a $T \sim 40$ years e-fold time, corresponding a hypothesized anthropogenic transient (somehow tied to human activity, climate, or CO₂). Under this forcing, a box with, e.g., a 10-year turnover will produce a smaller isotopic flux (change in δ^* reservoir size) compared to a box with a 60 year turnover time. With this one-box model as a building block, one could then construct a two-box model, with the change in Δ being different for the two boxes (one box representing C₃, one C₄). One could also construct an otherwise identical two-box model in which the change in Δ is the same for both boxes. The relevant version of Eq. 6 would be derived by adjusting the change in Δ for the second version to match the isotopic flux, i.e. change in the sum of δ^* (reservoir size), for the first version. The proper weighting of C₃ and C₄ discrimination will depend on three time constants (two turnover times, and atmospheric time constant of 40 years), combined into two dimensional parameters.

To address differential turnover of C₃ and C₄ in connection with the comparison with Keeling et al (2017), the revised draft introduces a new concept of "atmospheric $\Delta^{13}\text{C}$ ". This concept needs a more careful definition. As I understand it, the concept refers to the effective change in discrimination required to explain trends in atmospheric $\delta^{13}\text{C}$ per Keeling et al (2017). It weights the discrimination in favor of photosynthesis which enters long-lived pools, because pools with rapid turnover do not contain much biomass and thus have minimal impact on the abundance of ¹³C/¹²C ratio of atmospheric CO₂. The two-box model which I described in the last paragraph might add clarity to what is meant by this term and how to verbalize it.

I think it's clear that the P model is incompatible with the sensitivity from Keeling et al (2017) in the limit that the turnover time of C₄ is infinitely fast compared to C₃. In this limit, the Keeling et al sensitivity of $0.014 \pm 0.007 \text{ ‰ ppm}^{-1}$ would apply to C₃ plants only. Although I'm a bit confused about the numbers (see below), the text suggests that the P model gets a smaller sensitivity for C₃ of 0.003 ‰ ppm^{-1} (line 163). The difference compared to the simple model of Keeling et al makes sense considering the differing assumptions about mesophyll effects. I think the paper needs to at least briefly address this limit and the incompatibility with Keeling et al in this limit.

The authors consider cases with different turnover rates of C₃ and C₄ carbon, including a case which assumes the turnover time of C₄ is ten time shorter than that of C₃ ($\tau_{\text{model}} = 0.1$). Shouldn't this case be essentially equivalent to assuming that C₄ turnover is infinitely fast (i.e. the case in previous paragraph)? According to the text, however, this case yields a sensitivity of 0.018 ‰ ppm^{-1} (line 177), which is much larger than the value of 0.003 ‰ ppm^{-1} (line 163). This may be another indication that there is something fishy with the handling of the differential turnover of C₄ and C₃.

I suspect there may be a similar issue with the treatment of differential turnover with respect to soil carbon, but the method used is not very well described. I assume Eq (6) might have been applied here? Again, I think there's an issue because of the fishy mathematical limits, as I mentioned above. The formula here should be simple because one is addressing a steady state with just two time constants (no 40-year transient).

I have trouble reconciling the trends shown in Figure 4e with numbers presented in the text. The green and purple curve(s) in Figure 4e, first panel (C₃) both increase by about 1 permil over 35 years, implying a rate of ~ 0.03 per mil per year. In contrast, the text (line 162) lists a rate of 0.005 permil per year, which is 6 times smaller. There would appear to be a similar problem the C₄ panel.

Perhaps a related issue is that the legend of Figure 4 is not clear. The main text implies Figure 4e is showing global averages for all C3 or C4. But the caption indicates that the green curve is "Natural grasslands" and the violet curve is for "Natural and crops" (whatever that means). Neither curve would appear to be showing a combined measure over all C3 or C4 plants, as referenced in the main text.

On rereading, I note that the basis of the ratio F4 is not defined. F4 appears to be some fractional measure of cover. But what? Above ground biomass fraction? Fractional light interception? How would this be defined in the field?

On rereading, I also note that *GPPC3,pot* and *GPPC4,pot* may not be defined.

Reviewer #2 (Remarks to the Author):

This reviewer was satisfied with the responses from the authors, especially as related to the impacts of environmental forcing on trends in GPP and $\delta^{13}C$ discrimination. This reviewer thanks the authors for their detailed responses. I still have some lingering minor questions/suggestions which are listed below:

Has the model been validated for the exact setup you are using in this manuscript including time period, spatial region, model parameterization and maybe most importantly; meteorological forcing? I realize the focus of this manuscript is how trends in C3/C4 transition impact $\delta^{13}C$, but trends in atmospheric forcing also play a role. The temporal trends in your meteorological forcing (Figure S5) show some conflicts (e.g. VPD and soil moisture). Although atmospheric CO₂ seems to be the main driver in the C3/C4 transition, it would also be interesting to show spatially explicit trends in VPD, temperature, and soil moisture to compare with trends in spatially explicit C3/C4 landcover. Figures S4 and S5 get close to this, but not exactly.

Figure 3a-3b: Given your legend and units and layout within this figure, it is almost impossible to visually detect any increasing C4 trends (green) in these figures.

Is the Figure S1 workflow correct? It claims the temperature forcing is coming from the WFDEI data product, however, the methods state the temperature is coming from CRU. The methods state all met forcing data is derived from CRU with the exception of the SWdown which comes from WFDEI. This needs some clarification.

"Figure S5 Global average temporal changes in (a) atmospheric CO₂ concentrations (ppm), (b) daytime air temperature (T_{air}, °C), (c) vapour pressure deficit (VPD, kPa) and (d) soil moisture (θ , m³ m⁻³)"

Define the meteorological forcing data (CRU and WRFDEI) here again for clarity. It is curious that the temperature trends are about steady whereas soil moisture is increasing and VPD is increasing. Soil moisture and VPD seem to show opposing trends, however, this may be because it is a seasonal and spatial average which hides some region specific behavior. Also could be because you are only showing photosynthetically active periods (as described in Methods) which is misleading when looking at decadal trends where general climate warming signal may not be fully captured. This should be clarified.

Figure S9 : What is the significance of the treecover > 58% threshold?

Communications Earth & Environment is committed to improving transparency in authorship. As part of our efforts in this direction, we are now requesting that all authors identified as 'corresponding author' create and link their Open Researcher and Contributor Identifier (ORCID) with their account on the Manuscript Tracking System prior to acceptance. ORCID helps the scientific community achieve unambiguous attribution of all scholarly contributions. You can create and link your ORCID from the home page of the Manuscript Tracking System by clicking on 'Modify my Springer Nature account' and following the instructions in the link below. Please also inform all co-authors that they can add their ORCIDs to their accounts and that they must do so prior to acceptance.

Version 2:

Decision Letter:

Dear Dr Lavergne,

First of all, please allow me to apologise sincerely for the long delay in sending a decision on your manuscript titled "The recent decline in C₄ vegetation abundance contributes only slightly to the increase in atmospheric carbon isotopic composition" - unfortunately we had to replace one of the reviewers at a late stage. The article has now been seen by our original reviewer 2, and a new reviewer 3 (replacing the original reviewer 2); their comments appear below. In light of their advice we are delighted to say that we are happy, in principle, to publish a suitably revised version in Communications Earth & Environment, provided you address the remaining points raised by reviewer 3.

We therefore invite you to revise your paper one last time. At the same time we ask that you edit your manuscript to comply with our format requirements and to maximise the accessibility and therefore the impact of your work.

EDITORIAL REQUESTS:

*****Please take care to match our formatting and policy requirements. We will check revised manuscript and return manuscripts that do not comply. Such requests will lead to delays. *****

SUBMISSION INFORMATION:

OPEN ACCESS:

Communications Earth & Environment is a fully open access journal. Articles are made freely accessible on publication. For further information about article processing charges, open access funding, and advice and support from Nature Portfolio, please visit <https://www.nature.com/commsenv/open-access>

Link Redacted

Best regards,

Heike Langenberg, PhD
Chief Editor
Communications Earth & Environment
Communications Sustainability

On Bluesky:
@commsearth.nature.com
@commssustain.nature.com

REVIEWERS' COMMENTS:

Reviewer #2 (Remarks to the Author):

The authors response to my remaining questions were acceptable.

Reviewer #3 (Remarks to the Author):

This manuscript focused on a scientific question that was argued in different references : Does the C3/C4 abundance influence the isotope composition of atmosphere? There exhibited an obvious innovation and this work will be benefit for an improvement of the carbon cycling under the background of global warming. However, there are some concerns for the method part. First, the manuscript classified the plants to C3 and C4, is it possible that the CAM plants disturbed the results? Secondly, the estimation of isotope composition of soil was not well presented. For instance, in line 431, does this estimation consider the isotopic fractionation in soil process ? Thirdly, a table describing the detail of remote sensing data including the resolution, duration and source was recommended. Fourth, although the manuscript suggested that the changes in C3/C4 did not fully explain the isotope compositions of the CO₂, the possible reasons should be described more specifically.

Some specific comments are as follows:

- (1) Line 38 : "weaker" is not a suitable word in the sentence.
- (2) The link between GPP and delta C should better be described to enhance the logics in the introduction.
- (3) Some dataset like VPD and temperature may also produce some uncertainties. I recommended that you could emphasize the reliability of this products.
- (4) Actually, the isotope composition of CO₂, leaves, and soil all changed with seasons. Dose this variation influence your evaluation? Maybe a short discussion will helpful for a better understanding of the seasonal effects.

** Visit Nature Portfolio's author and referees' website at www.nature.com/authors for information about policies, services and author benefits**

We thank the two reviewers for their thorough review of our manuscript. Their suggestions have significantly contributed to improve the manuscript. Below in red you can find our responses to their comments.

Reviewer #1 (Remarks to the Author):

This study, which builds upon Lavergne et al (2022), presents results of a model (P model) that estimates the distribution of C3 and C4 vegetation, allowing assessments of changes in the C3/C4 distribution and impacts on GPP and isotopic discrimination. The model predicts that rising CO₂ has driven a decrease in C4, which drives an upwards trend in global discrimination and modifies the trajectory in GPP. The discrimination increase agrees well with an independent estimate of the trend in global discrimination based on the atmospheric $\delta^{13}\text{C}$ trend from Keeling et al. (2017).

The paper is overall well written, makes some interesting points, and is useful in presenting detailed results from the P model. The main issue is that it overlooks some important caveats.

We thank the reviewer for pointing out some caveats to our analyses and approach. We have tried to address them all in our revisions.

One caveat involves the comparison with the Keeling et al (2017). A change in discrimination can impact the atmospheric $\delta^{13}\text{C}$ trend **only to the extent that it isotopically alters carbon stored in vegetation and soils**. The land biospheric impact on the global $\delta^{13}\text{C}$ budget thus involves a convolution of discrimination AND carbon turnover. Photosynthesis which feeds a short-lived pool has less leverage than photosynthesis which fuels a long-lived pool. In this context, it's highly relevant that the **turnover of C4 carbon is generally more rapid than turnover of C3 carbon, as C4 is rare in trees**. The model used in Keeling et al did not address differential turnover of C3 versus C4, but the issue was addressed briefly in discussion, where the impact of C4 was discounted. In any case, the discrimination trend estimated in Keeling et al (2017) should not be naively interpreted as a GPP-weighted average of C3 and C4 discrimination. Rather, it must be weighted towards C3, possibly quite heavily. Clearly more work is needed on this topic, e.g. one needs a model that includes carbon turnover in a range of pools divided by C3 and C4. For now, the authors need to at least point out this important caveat.

We agree with the reviewer that a direct comparison of GPP-weighted average of C₃ and C₄ carbon isotopic discrimination ($\Delta^{13}\text{C}$) and atmospheric $\Delta^{13}\text{C}$ may be an oversimplification as the carbon turnover of C₃ and C₄ woody biomass should also be accounted for. We have now tried to weight the contributions of C₃ and C₄ plants to atmospheric $\Delta^{13}\text{C}$ via their different turnover rates using a simple approach. We assumed one carbon pool representing land and used different carbon residence times for C₄ and C₃ plants. We have clarified our approach to take into account carbon turnovers in the Methods section (L339-348). Since this approach is still a simplification and does not account for more carbon pools, we have also discussed the limitations and caveats of the approach in the Discussion section (L296-300).

L339-348: 'Since the carbon turnover of C₄ plants is faster than that of C₃ plants, an estimate of atmospheric Δ (Δ_{atm}) should be weighted considering these different turnover rates. Here we calculate Δ_{atm} considering only one land carbon pool as:

$$\Delta_{atm} = F_{4,tot} \sum_t \frac{\Delta_{4,t} \times GPP_{4,t}}{\sum_t GPP_{4,t}} \tau_{ratio} + (1 - F_{4,tot}) \sum_t \frac{\Delta_{3,t} \times GPP_{3,t}}{\sum_t GPP_{3,t}} \quad (6)$$

τ_{ratio} indicates the ratio of carbon turnover rates of C₃ and C₄ plants. Since biomes dominated by C₄ plants have higher carbon turnover rates than those dominated by C₃ plants^{42,43} and C₄-derived soil organic carbon tends to decompose twice faster than its C₃ counterpart in mixed C₃/C₄ soils⁹, for simplicity, we assume that carbon turnover rates in C₄ plants are two to ten times higher than those in C₃ plants to span a range of plausible values^{42,43} (τ_{ratio} ranging between 0.1 and 0.5), leading to lower contribution of C₄ than C₃ plants to Δ_{atm}.

L296-300: 'We recognize that our approach to incorporating the effect of different carbon turnover rates for C₄ and C₃ plants into our Δ¹³C_{atm} estimates is too simple as it only considers one land carbon pool, nevertheless it shows that the higher the turnover of C₄ carbon, the lower the contribution of C₄ photosynthesis to Δ¹³C_{atm}, consistent with Keeling et al. (2017)² hypothesis.'

An accounting of differences in the turnover of C₃ versus C₄ carbon is also needed to support the comparisons that are presented between the P model and measurements of δ¹³C in soil organic matter (Figure 1). How was this addressed? If this was not addressed, some caveats are needed. See Wynn and Bird (2007, <https://doi.org/10.1111/j.1365-2486.2007.01435.x>).

We also acknowledge that differences in the carbon turnover of C₃ versus C₄ plants should be accounted for when comparing predicted and measured soil organic matter δ¹³C. We have weighted predicted δ¹³C values with the respective carbon turnover rates (see Methods L386-388) and discussed the limitations of our approach in the Discussion section.

L386-388 'Predicted Δ¹³C values for C₃ and C₄ plants were converted to δ¹³C using atmospheric δ¹³CO₂ from Graven et al. (2017)⁴⁹ and weighted by their relative fraction and carbon turnover rates to enable comparisons with the observed network.'

An important caveat on the use of tree rings to infer long-term trends in discrimination is that these studies (to my knowledge) have not yet addressed a range of complications that parallel those that arise using tree rings to infer long-term trends in growth (Brienen et al, doi: 10.1111/gcb.13605).

The problems highlighted by Brienen et al. (2017) are mainly related to long-term growth trends, and to our knowledge should not impact δ¹³C and Δ¹³C trends. Nevertheless, we have mentioned potential issues with using tree-ring isotopic data to infer Δ¹³C trends in the Introduction section.

L50-55 'It is possible that post-photosynthetic fractionation processes^{3,8} and intrinsic age-related changes in tree development over their lifespan such as tree height^{6,7} affect inferences of long-term Δ¹³C trends from tree rings, explaining the discrepancies between atmospheric and plant Δ¹³C estimates. However, these effects are still not well understood and quantified, and no alternative data for C₃ plants is currently available.'

Some further discussion is merited on mesophyll impacts on discrimination. The current version of the p-model neglects mesophyll impacts on discrimination, which the authors justify based on the Lavergne (2022) et al study suggesting that mesophyll impacts have little impact on the discrimination in response to rising CO₂. As far as I can tell, the Lavergne (2022) result is tied to a built-in assumption that mesophyll conductance tends to scale in proportion to stomatal conductance in response to rising CO₂. This is an interesting hypothesis, but is this really a settled issue? What are the bounds in its validity? It would seem appropriate to add the caveat that the uncertainty arising from this assumption has not been addressed, and suggest further work to reduce these uncertainties.

Maybe the most efficient way to address these caveats is to add a section at the end on needed follow-up studies.

We have expanded on the limitations of our assumption regarding the mesophyll impact on carbon isotopic discrimination and have mentioned potential follow-up studies required to address these limitations in the Discussion section.

L275-281: 'We acknowledge that this mesophyll effect estimates heavily relies on the assumption that the ratio of stomatal to mesophyll conductance is independent of environmental factors, leading to an optimal ratio of the chloroplastic to ambient CO₂²⁶. The uncertainty arising from this assumption has not been addressed yet, and further work involving field measurements of both stomatal and mesophyll conductances and stable carbon isotopes in leaves and wood in controlled environments could help quantify and reduce these uncertainties. '

Minor points:

49. Whether there is a global trend or not needs to be decided by observations not models. The jury is still out, I think.

We agree that global $\Delta^{13}\text{C}$ trends should be determined by observations rather than by models. However, we do think that models that incorporate a mechanistic understanding of the processes at play are complementary and can help understand $\Delta^{13}\text{C}$ trends.

Figure 1. It might help to put "observed" within Figure 1b, paralleling 1c and 1d.

Done

Figure 2. On 2b and 2c, add the word "from" as in "from Still2009". For 2a, also add "from..." specifying the model name.

Done

99: Higher than what?

We have corrected to 'high'.

Figure 3. Specify model and time period in all three panels.

Done

115: The basis of GPP calculation is not entirely clear from the wording. Per unit surface area?. Maybe this needs a more precise term than "GPP". The wording needs to allow that the reader may not yet have read the Method section.

We have corrected as: 'the predicted gross primary production (GPP) per unit surface area'.

Eq (1) and Eq. (2a) appear to use inconsistent notation $F_{4,nat}$ versus $F_4, natural$.

Corrected

301. spell out Ref 43 to complete the sentence.

Done

Reviewer #2 (Remarks to the Author):

The authors use the P light use efficiency model combined with a C3/C4 competition model to demonstrate that the recent increase (2001-2016) in natural C3 landcover is likely responsible for increased $\delta^{13}C$ photosynthesis globally and the reason for the decreasing trend of $\delta^{13}C$ atmosphere overall. As validation for their model approach, the reviewers compare their C3/C4 model and simulated GPP against Still and Luo approaches, claiming the Still approach significantly overestimates C4 landcover overall. They use site level $\delta^{13}C$ leaf and $\delta^{13}C$ soil estimates as validation to their model approach.

Although this reviewer found the author's C3/C4 landcover mechanism a plausible explanation for the decreased trend in $\delta^{13}C$ atmosphere there were two significant concerns.

We thank the reviewer for their thorough review of our paper.

First the validation of the P model and C3/C4 competition model was very limited. It would have been more compelling if the P model was validated against GPP measurements at flux tower sites or against global reanalysis products like FLUXCOM (Jung et al.). The author's do supply Figure 1a as a type of validation for the photosynthetic discrimination model, however, it does seem like the model underestimates the magnitude of C3 discrimination when compared to the leaf observations, and thus might be compensating for this bias with increased C3 landcover.

The P model predictions of GPP have been extensively validated against eddy-covariance flux data in other studies [see for example Stocker et al. 2020]. Cai & Prentice (2020) ERL even showed that it can reproduce trends in GPP at flux sites with long records. Re-validating GPP using similar flux data in the present study would not provide new evidence of the validity of the model. We have now clarified in the Supplementary Material that model validation of GPP can be found in the previous studies.

We do not think validating GPP predictions with FLUXCOM is appropriate as this product do not contain temporal trend in GPP linked to CO₂ [see O'Sullivan et al. 2020; Walker et al. 2020; Jung et al. 2020; Ruehr et al. 2023 Nat Rev Earth Environ].

We agree that the model underestimates the variability of $\Delta^{13}\text{C}$ in C₃ plants compared to the leaf observations (standard deviation equal to 1.56 compared to 2.63‰, respectively), and that this underestimation may affect global $\Delta^{13}\text{C}$. We have now acknowledged this bias in the Results section (L115-119).

L115-119: 'The skills of the model to predict $\Delta^{13}\text{C}$ for C₃ and C₄ plants were reasonably good (coefficients of determination $R^2 = 0.50, 0.23$ and 0.92 , respectively for C₃, C₄ and total (C₃ and C₄) plants; Figure 1a) despite the model underestimating the leaf-derived variability of $\Delta^{13}\text{C}$ (standard deviation = 1.56‰ versus 2.63‰ , respectively, for C₃ plants and, 0.60‰ versus 1.32‰ , respectively, for C₄ plants).'

Cai, W. & Prentice, I. C. Recent trends in gross primary production and their drivers: analysis and modelling at flux-sites and global scales. *Environ. Res. Lett.* **15**, 124050 (2020).

Stocker, B. D. et al. P-model v1.0: an optimality-based light use efficiency model for simulating ecosystem gross primary production. *Geosci. Model Dev.* **13**, 1545–1581 (2020).

O'Sullivan, M., et al. Climate-driven variability and trends in plant productivity over recent decades based on three global products. *Global Biogeochemical Cycles*, **34**, e2020GB006613 (2020). <https://doi.org/10.1029/2020GB006613>

Walker, A. P. et al. Integrating the evidence for a terrestrial carbon sink caused by increasing atmospheric CO₂. *New Phytol.* **229**, 2413–2445 (2021).

Jung, M. et al. Scaling carbon fluxes from eddy covariance sites to globe: synthesis and evaluation of the FLUXCOM approach. *Biogeosciences* **17**, 1343–1365 (2020).

Ruehr, S., Keenan, T.F., Williams, C. et al. Evidence and attribution of the enhanced land carbon sink. *Nat Rev Earth Environ* **4**, 518–534 (2023). <https://doi.org/10.1038/s43017-023-00456-3>

Second, the authors do not discuss to what extent trends in VPD (which influence discrimination) may have impacted the atmospheric signature. To what extent did your meteorological forcing product show trends in VPD and soil moisture and how did this influence the discrimination in the P model? The authors provide a brief discussion of model GPP sensitivity to environmental variables in the supplement, but the sensitivity of the model photosynthetic discrimination is lacking.

We thank the reviewer for pointing out the potential significant impact of VPD on $\Delta^{13}\text{C}$ and its atmospheric signature. We have now added a sensitivity analysis of $\Delta^{13}\text{C}$ for C₃ and C₄ plants to their environmental drivers in Figure S10 (now S12) in the same way as for GPP (see below) and have described the analysis in Text S2.

Both $\Delta^{13}\text{C}_{\text{C}_3}$ and $\Delta^{13}\text{C}_{\text{C}_4}$ depend on the ratio of leaf intercellular to ambient CO₂ (c_i/c_a). However, while $\Delta^{13}\text{C}_{\text{C}_3}$ tends to increase with c_i/c_a (Farquhar et al., 1982), $\Delta^{13}\text{C}_{\text{C}_4}$ decreases (Farquhar, 1983; von Caemmerer et al., 2014 - see also Figure S10 below). As a result, $\Delta^{13}\text{C}_{\text{C}_3}$ and $\Delta^{13}\text{C}_{\text{C}_4}$ respond inversely to the environmental drivers of c_i/c_a , i.e., $\Delta^{13}\text{C}_{\text{C}_3}$

increases with higher T_{air} and soil moisture θ , but decreases with high VPD and atmospheric pressure as indicated by elevation z (see below).

Farquhar, G. D., O'Leary, M. H., & Berry, J. A. (1982). On the Relationship between Carbon Isotope Discrimination and the Intercellular Carbon Dioxide Concentration in Leaves. *Australian Journal of Plant Physiology*, 9, 121-137. <https://doi.org/10.1071/PP9820121>

Farquhar, G. (1983). On the nature of carbon isotope discrimination in C4 species. *Aust. J. Plant Physiol.*, 10, 205-226.

von Caemmerer, S., Ghannoum, O., Pengelly, J. J., & Cousins, A. B. (2014, Jul). Carbon isotope discrimination as a tool to explore C4 photosynthesis. *J Exp Bot*, 65(13), 3459-3470. <https://doi.org/10.1093/jxb/eru127>

Figure S10

Figure S12

Detailed comments below:

Abstract:

"We conclude that the magnitude of the decrease in global atmospheric $\delta^{13}\text{CO}_2$ can be partly explained by global changes in the distribution of C3/C4 plants."

This implies that C4 natural grasslands are strongly decreasing. Does it make sense that the CO2 concentration increase leads to this given increased aridity which favors C4 plants?

In the paper we show that despite the increase in C4 crops over the recent decades, the fraction of C4 natural grassland has strongly decreased. While warmer environments may favour C4 plants, increasing CO2 concentrations in turn tend to benefit C3 plants. Note that since photorespiration in C4 plants is greatly reduced compared to that in C3 plants, we have assumed that it can be neglected for simplicity. As a result, C4 photosynthesis predicted by the P model does not vary with VPD (see also Text S1 and Equation 1 in Supplementary Material and Figure S12 above). Nevertheless, we recognize that C4 photosynthesis can be inhibited by low soil moisture.

To highlight the higher impact of CO2 versus dry/warm conditions on C4 fraction, we have now added a new attribution analysis (see Methods). By accounting for the differing effects of T_{air} , VPD, CO2 and soil moisture θ on C3 and C4 photosynthesis, we show that the impact of CO2 on the fraction of C4 plants is stronger than that of T_{air} or VPD (see below new Figure 5a-b). This result is in line with Luo et al (2024) study (see below their figure).

New Figure 4a-b (absolute F_4 change as a response to the four environmental drivers)

From Luo et al. (2024)

Luo, X. et al. Mapping the global distribution of C4 vegetation using observations and optimality theory. Nat. Commun. 15, 1219 (2024).

Introduction:

"A more recent work has incorporated C3/C4 adaptation and acclimation to recent

environmental changes based on an optimality model and observations. The derived map suggests that the global fraction of C₄ plants has decreased over 2001-2019 period due to a decrease in C₄ natural grasses with elevated CO₂, even though the abundance of C₄ crops has increased.”

I feel like there needs to be a bit more explanation of why C₃ plants discriminate more against $\delta^{13}\text{C}$ as compared to C₄ plants. More background. No explanation of alternative hypotheses to C₃/C₄ land surface transition. See Raczka et al., 2017 JGR-Biogeosciences.

It is widely known that C₃ plants are depleted in ¹³C compared to C₄ plants [see for example O’Leary, 1981]. However, to provide more context, we have added a sentence in the Introduction explaining the reasons for this pattern.

L72-80 ‘Variations in $\Delta^{13}\text{C}$ are closely related to environmental-driven changes in the stomatal limitation of photosynthesis (i.e. the ratio of leaf internal to ambient partial pressure of CO₂), but also depend on the pathway of carbon assimilation. Isotopic fractionation during the diffusion of CO₂ through the stomata primarily influences $\Delta^{13}\text{C}$ in C₄ plants, while fractionation during Rubisco carboxylation has a stronger imprint on $\Delta^{13}\text{C}$ in C₃ plants, resulting in C₃ plants being depleted in ¹³C compared to C₄ plants^{16–18}. Knowledge of the different isotopic signatures of C₃ and C₄ photosynthetic pathways and of their relative coverage across the globe can be used to estimate average $\delta^{13}\text{C}$ across terrestrial environments and so global $\Delta^{13}\text{C}$.’

We are unsure what the reviewer is referring to when stating ‘No explanation of alternative hypotheses to C₃/C₄ land surface transition’. The goal of our study is to determine whether recent changes in C₃/C₄ distribution and abundance influence global land and atmospheric $\Delta^{13}\text{C}$. We are proposing a new simple model of C₃/C₄ changes to test such an effect and are comparing it to the model recently developed by Luo et al. (2024).

O’Leary, M. H. Carbon isotope fractionation in plants. *Phytochemistry* 20, 553–567 (1981).

Also, I feel there is a lack of background given to post-photosynthetic fractionation processes in general, that could influence $\delta^{13}\text{C}$ soil (See Bruggemann et al., 2011)

We agree that post-photosynthetic fractionation processes influence $\delta^{13}\text{C}$ in plants and soils, however, those effects are still not well understood and quantified [see also Introduction/Discussion in our previous publication - Lavergne et al. 2022 GCB]. We have added a sentence in the Introduction to provide some background regarding post-photosynthetic fractionation processes.

L50-55 ‘It is possible that post-photosynthetic fractionation processes^{3,6} and intrinsic age-related changes in tree development over their lifespan such as tree height^{7,8} affect inferences of long-term $\Delta^{13}\text{C}$ trends from tree rings, explaining the discrepancies between atmospheric and plant $\Delta^{13}\text{C}$ estimates. However, these effects are still not well understood and quantified, and no alternative data for C₃ plants is currently available.’

Lavergne, A., Hemming, D., Prentice, I. C., Guerrieri, R., Oliver, R. J., & Graven, H. (2022). Global decadal variability of plant carbon isotope discrimination and its link to gross primary production. *Glob Chang Biol*, 28(2), 524-541. <https://doi.org/10.1111/gcb.15924>

Methods:

Authors evaluated the light-use efficiency (P model) against leaf isotope data for C3 and C4 plants, but what about GPP and NPP predictions based on flux tower data, NEON site data or global GPP products like FLUXCOM? Has the carbon model within the p model been evaluated/validated at all? You are looking at delphoto, and not delland atmosphere exchange?

As stated above, GPP predicted by the P model has been widely validated using eddy-covariance flux data in other recent studies [see for example Stocker et al. 2020, GMD and Cai & Prentice 2020, ERL]. We therefore decided not to show the GPP model validation in the present study. Here, we predict leaf $\Delta^{13}\text{C}$ and associated land $\Delta^{13}\text{C}$ and compare them to atmospheric $\Delta^{13}\text{C}$ trend estimated by Keeling et al. (2017). As pointed out by Reviewer #1, we need to consider differences in carbon turnover between C₃ and C₄ plants to be able to make a full comparison of land $\Delta^{13}\text{C}$ with atmospheric $\Delta^{13}\text{C}$. We have implemented a simple formulation to include such an effect in the revised manuscript but are also acknowledging the simplification of our approach.

Keeling, R. F. et al. Atmospheric evidence for a global secular increase in carbon isotopic discrimination of land photosynthesis. *Proc Natl Acad Sci U A* 114, 10361–10366 (2017).

Results:

Figure 3: I don't understand how F4 trends can be decreasing everywhere, all across the globe. Especially when the authors state F4 is increasing over agriculture areas. This assumes the effect of CO₂ fertilization is superseding any drying/warming trends everywhere. Seems highly unlikely.

As Figure 3a-b shows, F_4 including both natural grasslands and crops is decreasing in many regions but not everywhere. F_4 increase in equatorial regions and in some high latitudes of the Northern Hemisphere (see below in green). Therefore, the effect of CO₂ fertilization on C₃ plants is superseding the effect of warming/drying on C₄ plants in many regions but not everywhere.

(a) Our model over 1982-2016

(b) Our model over 2001-2016

Figure 3a-b

Discussion:

The only validation has been with sporadic site level leaf $\delta^{13}\text{C}$ for C3 and C4 species (Fig1). When the C3 and C4 species are considered individually the skill is quite modest, but it looks like they are comparing coarse grid cell (0.5x0.5) against site level data. The spatial mismatch is significant and the meteorological forcing mismatch impedes a fair comparison.

We recognize that using a relatively coarse grid cell to predict $\Delta^{13}\text{C}$ may impact the comparison between predicted and leaf-derived $\Delta^{13}\text{C}$ values. However, the aim of our study is to examine global spatial patterns and trends and we believe that the choice of spatial resolution for the climate records is a compromise between spatial accuracy and the goal of our study. We note that the lack of weather stations close to the study sites from which the data are derived prevents us from running the model with local climate data for these specific sites. Wang et al. (2017) already carried out an extensive validation of leaf c_i/c_a ratios (based on $\delta^{13}\text{C}$ measurements) at sites around the world. Here we used soil data because they can inform about the overall discrimination by the plant community, as opposed to an individual plants species.

Wang et al. (2017) Nature Plants

We discussed the potential bias of using a 0.5x0.5 grid in validating our model in the Methods (L389-392).

L389-392: 'We acknowledge that a relatively coarse grid cell to predict $\Delta^{13}\text{C}$ may impact the comparison between predicted and leaf-derived $\Delta^{13}\text{C}$ values. However, the lack of weather stations close to the study sites from which the data are derived prevents us from running the model with local climate data for these specific sites.'

P Optimality Model: Figure S1 attempts to explain how it is constrained. The Paruelo based logistic regression looks poorly fit to the observations. There is no mention how the uncertainty between Adv4 and Sh4 contribute to uncertainty in the prediction.

The share in C₄ plants (Sh4) model developed based on a logistic regression of the Paruelo dataset with the predicted advantage of C₄ over C₃ plants (Adv4) only explains 29% of the variability of the data. We agree that the fit could have been higher. We did a thorough search in the literature to find additional Sh4 data for the observed contribution of C₄ plants to total ecosystem GPP without success. Without such dataset, the relationship cannot be refined. Despite this limitation, our logistic prediction shows that Sh4 increases with increasing Adv4, which is what we would expect from the theory.

We have now discussed in the Supplementary Material how the uncertainties in Adv4 and Sh4 could contribute to uncertainties in the predictions of the fraction of C₄ plants and have added a 95% confidence interval on the regression in Figure S8 (see below).

Figure S8

The authors claim that increases in CO₂ are allowing for C₃ species to outcompete C₄ species, however, the authors do not discuss the impact of VPD and soil moisture have on C₃/C₄ competition. Warmer, drier conditions tends to favor C₄ species. Furthermore the authors show their P optimality model is highly sensitive to VPD (much more so than CO₂) in terms of GPP – these implies stomatal conductance is strongly reduced which should increase discrimination in C₃ species. Could simply a stable C₃/C₄ spatial map, combined with increased discrimination of C₃ species account for trends in atmospheric del13C? (No, Lavergne tried this in a previous manuscript, but could this be because of underestimated C₃ discrimination?)

We apologize for the confusion here. While Figure S10a-b (now S12) shows the sensitivity of GPP for C₃ and C₄ plants to their environmental drivers, the magnitudes of the standardized coefficients are not actually representing their overall contributions but only the direction of their effects. Cai and Prentice (2020) already investigated the contribution of environmental variables to predicted GPP change over the same period 1982-2016, showing that globally, increasing CO₂ concentrations are the most important drivers of GPP increase, followed by T_{air}, fAPAR, PPFD and to a lower extent VPD and soil moisture (see below).

When comparing simulations of GPP for four scenarios (when either CO₂, T_{air}, VPD or soil moisture is assumed constant over the whole period) with the simulation including all effects, (see new attribution analysis and new Figure 5c-e below), it is clear that CO₂ is the main driver of total GPP and that T_{air} and VPD have a minor effect.

Our results point towards a stronger effect of high CO₂ concentrations on C₃ plants than of drying and warming on C₄ plants, resulting in a decrease in C₄ fraction in most (but not all)

regions across the globe. The different spatial impacts of CO_2 , T_{air} , VPD or soil moisture on F_4 are also highlighted in new Figure S4.

Note that there was a mistake in the title of the original Figure S11 which shows the sensitivity of F_4 to CO_2 , VPD and soil moisture but not of GPP as erroneously suggested. Since the figure does not actually reflect the contributions of each driver to F_4 , we have now replaced it with the new Figure 5.

It is now acknowledged that C_3/C_4 distribution and abundance varies over time because of environmental changes (Luo et al., 2024), so assuming a constant C_3/C_4 spatial map is not realistic. When considering constant C_3/C_4 distribution, total $\Delta^{13}\text{C}$ increases only slightly and could be considered negligible (as also shown in our previous publication, Lavergne et al. 2022 GCB). However, when we consider the decrease in C_4 fraction, total $\Delta^{13}\text{C}$ increases more strongly. Reviewer #1 suggested that we should account for the different carbon turnover rates of C_3 and C_4 plants to estimate global atmospheric $\Delta^{13}\text{C}$ as they are higher in C_4 than C_3 plants. While accounting for carbon turnover by weighting our predictions of total $\Delta^{13}\text{C}$ using carbon turnover rates estimates for C_3 and C_4 plants affect the mean predicted $\Delta^{13}\text{C}$ values, it does not significantly impact the resulting trends.

Cai and Prentice (2020)

Figure 5c-e

The fact that their C_3/C_4 competition model is showing increased C_3 coverage with time *everywhere* across the globe, with a model that is way more sensitive to VPD, than CO_2 (Figure S11), indicates to me that the met forcing product they used is showing stable or increased moisture across this period. Did they check this for trends? They need more investigation into their met forcing product. I think more explanation of the C_3 fractionation results and sensitivity to VPD is necessary. To show the sensitivity of C_4 plants to c_i/c_a (Figure S8), but no mention of sensitivity of C_3 discrimination to environmental drivers seems odd.

As stated above, C_4 coverage decreases in most regions of the world (but not all).

The original Figure S11 does not actually reflect the contributions of each environmental driver to F_4 but the new Figure 5 does. The contribution of each individual effect varies over time (Fig 5a) and, at least over the last years, F_4 is more strongly influenced by CO_2 than by T_{air} (Fig 5b).

As stated above, we have added a sensitivity analysis of $\Delta^{13}\text{C}$ to environmental drivers in the same way as GPP (Figure S12c-d) and have determined the contribution of CO_2 , T_{air} , VPD and soil moisture to GPP and $\Delta^{13}\text{C}$ in Figure 5.

Figure S8 (now S10) represents the linear regression used to predict $\Delta^{13}\text{C}$ for C_4 plants as a function of c_i/c_a values based on leaf-derived $\Delta^{13}\text{C}$ observations (see equation 12 in Text S1). In the previous version of the paper, we did not show similar plot for C_3 plants as $\Delta^{13}\text{C}$ for C_3 plants is predicted using equation 11 in Text S1. However, to acknowledge the reviewer's comment and show how $\Delta^{13}\text{C}$ for C_3 plants varies with c_i/c_a , we have now added this plot in Figure S10.

Why wouldn't you show sensitivity of C_3 and C_4 to $\delta^{13}\text{C}$ discrimination based on CO_2 , VPD, and soil moisture as well (Figure S11)? The authors show sensitivity to GPP, but don't actually validate the model at all against GPP site level or regional level (FLUXCOM) products.

As stated above, we have now added panels (c) and (d) in Figure S10 (now S12) to show the environmental sensitivity of $\Delta^{13}\text{C}$ for C_3 and C_4 plants to their environmental drivers.

There was a mistake in the original title of Figure S11 – the figure shows the sensitivity of F_4 (not GPP) to CO_2 , VPD and soil moisture. We have replaced Figure S11 by a new Figure 5 to show these contributions.

They show limited $\delta^{13}\text{C}$ leaf and $\delta^{13}\text{C}$ soil, but seem to focus on the $\delta^{13}\text{C}$ soil results which are less related to the photosynthetic fractionation than leaf $\delta^{13}\text{C}$. The $\delta^{13}\text{C}$ of soil is a function of $\delta^{13}\text{C}$ photosynthesis and $\delta^{13}\text{C}$ respiration and other post fractionation processes, which the P model does not include (Bruggemann et al., 2011)

Since post-photosynthetic fractionation processes are not yet well constrained and quantified, especially potential differences between C_3 and C_4 plants, we do not include them in the calculations but acknowledge their importance in the Discussion section.

$\delta^{13}\text{C}$ in plants and soils can be used to document changes in the abundance of C_3 and C_4 plants. $\delta^{13}\text{C}$ measured in soil organic matter reflects the isotopic composition of plants that grew at a given location before they decomposed, because $\Delta^{13}\text{C}$ depends on the carbon assimilation pathway. $\Delta^{13}\text{C}$ in C_3 species is primarily caused by fractionation during the diffusion of CO_2 from the atmosphere to the chloroplast, Rubisco carboxylation, and photorespiration (Farquhar et al., 1982). $\Delta^{13}\text{C}$ in C_4 species reflects biochemical fractionations by both Rubisco and phosphoenolpyruvate carboxylase (Farquhar, 1983). These isotopic differences between photosynthetic pathways can be exploited to evaluate models of C_3/C_4 competition using $\delta^{13}\text{C}$ measured in soil organic matter.

Reviewer #1 suggested that differences in the carbon turnover of C_3 versus C_4 plants should be accounted for when comparing predicted $\delta^{13}\text{C}$ and measured soil organic matter $\delta^{13}\text{C}$. Following this suggestion, we have now weighted predicted $\delta^{13}\text{C}$ values with the respective

carbon turnover rates. By doing so, the agreement between predicted and soil measured $\delta^{13}\text{C}$ slightly increased ($R^2 = 0.56$ instead of 0.54 using the original version; Figure 1).

Line. 161: "However, Still2009 tends to overestimate F4 in sub-Saharan and southern Africa and northern Australia compared to our map and that of Luo2024 - as also indicated by the higher predicted than observed $\delta^{13}\text{C}_{\text{soil}}$ values found in Still2009 for these regions (Figure 1c)."

But isn't $\delta^{13}\text{C}_{\text{leaf}}$ a better indicator of fractionation from photosynthesis as compared to $\delta^{13}\text{C}_{\text{soil}}$, which could be influenced by post-photosynthetic fractionation processes? Could you not show Figure 1A compared to Still and Luo?

As explained above, $\delta^{13}\text{C}_{\text{soil}}$ is a good indicator of changes in the abundance of C_3 and C_4 plants in a given site because of the different isotopic signatures of C_3 and C_4 photosynthetic pathways. This is why we are comparing predicted $\delta^{13}\text{C}$ weighted by the relative fraction and carbon turnover of C_3 and C_4 plants to $\delta^{13}\text{C}_{\text{soil}}$.

Line 197: "Nevertheless, results from our sensitivity analysis suggest similar patterns of variability of F4 with elevated CO_2 and water stress as with Luo2024 map, i.e., small negative effect of elevated CO_2 and positive effect of high VPD and low soil moisture on F4 (Text S2 and Figure S11)"

I don't understand this. Figure S11 shows dominant impact of VPD on GPP and likely discrimination, which would seem to allow C_4 species to outcompete C_3 . The authors suggest the opposite in the intro, that trends in CO_2 are overwhelming the F4 trend.

We apologize for the confusion. As stated above, the title of original Figure S11 was wrong. It should have been 'Sensitivity of F_4 to CO_2 , VPD and soil moisture'. Since the figure does not properly highlight the contributions of each environmental driver, we have now replaced it with new Figure 5 (see comments above).

Line 222: Have you demonstrated the simulated $\delta^{13}\text{C}$ for C_3 species is accurate? Has it been validated at the site level or any type of observation that matches the spatial scale of your simulation?

Since not all environmental variables needed to run the P model at the local scale are available at our study sites, we only compare predicted and leaf-derived $\Delta^{13}\text{C}$ for C_3 plants using the 0.5x0.5 grid data (see Figure 1a). Despite the potential spatial mismatch, predicted $\Delta^{13}\text{C}$ considering both C_3 and C_4 plants explains around 90% of the variability in leaf-derived $\Delta^{13}\text{C}$ observations.

Line 247: "Our study highlights the importance of considering recent C_3 and C_4 land cover changes with elevated atmospheric CO_2 and increasing water stress in the terrestrial carbon budget and pave the way for an improved evaluation of the mechanisms at play."

These authors did not consider/discuss the impact of water stress on the trends in GPP and $\delta^{13}\text{C}$. This was surprising considering the high sensitivity of the P model to VPD, and

suggests drought tolerant species C4, could maintain or perhaps expand their extend in a warmer/drier world. The authors need to discuss the pattern of VPD trends by region and over time, and how that impacted their results.

As stated above, we have now shown the real contributions of CO₂, T_{air}, VPD and soil moisture to GPP, Δ¹³C and F₄ by adding a new attribution study. We also corrected the sentence to better reflect our findings.

We thank the two reviewers for reviewing our manuscript for a second time. We appreciate that they both recognize our effort to address their initial concerns. Below in red you can find our responses to their comments.

Reviewer #1 (Remarks to the Author):

I laud the work done to address the issue of differences in turnover time between C3 and C4 carbon.

I note, however, that Eq. 6 looks fishy on mathematical grounds. A reality check is that Δ_{atm} remains bounded between the pure C4 and C3 limits as τ grades from zero to infinity. But according to Eq. (6), if τ is infinite, then Δ is infinite. And $\tau=0$ also misses the pure C3 limit.

We agree with the reviewer on the need to clarify the limits in equation 6. In the previous version of the manuscript, mathematically τ could grade from zero to infinity, however, it should have been bounded between 0 ($\Delta^{13}\text{C}$ completely dominated by Δ_{C3}) and a finite high value (where $\Delta^{13}\text{C}$ is dominated by Δ_{C4}). We should have added the following in the text:

If $\tau = 0$, $\Delta^{13}\text{C} = \Delta_{\text{C3}}$.

If $\tau = 1$, $\Delta^{13}\text{C} = \Delta_{\text{C4}} + \Delta_{\text{C3}}$.

If $\tau = 1000$, $\Delta^{13}\text{C} = 1000 * \Delta_{\text{C4}} + \Delta_{\text{C3}} \sim 1000 * \Delta_{\text{C4}}$.

Nevertheless, we acknowledge the oversimplification of our previous approach. In the revised manuscript, we have decided to use a more robust approach to determine the impact of varying fractions in C₃ and C₄ plants on land carbon isotopic discrimination ($\Delta^{13}\text{C}$) and ultimately on the isotopic composition of atmospheric CO₂ ($\delta^{13}\text{CO}_2$) (see below).

I also note that timescale of the factors driving changes in discrimination (e.g. the timescale of the CO₂ rise, or climate change) will dictate which carbon pools (long-lived versus short-lived) are most important. Even a very simple treatment of differential C₄/C₃ turnover in the context of changing discrimination therefore needs to include information on how fast discrimination is changing. I would be tempted to start by trying to understand the behavior of a one-box carbon model with exponentially changing discrimination ($\Delta = a + b e^{(t/T)}$) with a $T \sim 40$ years e-fold time, corresponding a hypothesized anthropogenic transient (somehow tied to human activity, climate, or CO₂). Under this forcing, a box with, e.g., a 10-year turnover will produce a smaller isotopic flux (change in δ^* reservoir size) compared to a box with a 60 year turnover time. With this one-box model as a building block, one could then construct a two-box model, with the change in Δ being different for the two boxes (one box representing C₃, one C₄). One could also construct an otherwise identical two-box model in which the change in Δ is the same for both boxes. The relevant version of Eq. 6 would be derived by adjusting the change in Δ for the second version to match the isotopic flux, i.e. change in the sum of δ^* (reservoir size), for the first version. The proper weighting of C₃ and C₄ discrimination will depend on three time constants (two turnover times, and atmospheric time constant of 40 years), combined into two dimensional parameters.

To address differential turnover of C₃ and C₄ in connection with the comparison with Keeling et al (2017), the revised draft introduces a new concept of "atmospheric $\Delta^{13}\text{C}$ ". This concept

needs a more careful definition. As I understand it, the concept refers to the effective change in discrimination required to explain trends in atmospheric $\delta^{13}\text{C}$ per Keeling et al (2017). It weights the discrimination in favor of photosynthesis which enters long-lived pools, because pools with rapid turnover do not contain much biomass and thus have minimal impact on the abundance of $^{13}\text{C}/^{12}\text{C}$ ratio of atmospheric CO_2 . The two-box model which I described in the last paragraph might add clarity to what is meant by this term and how to verbalize it.

We thank the reviewer for their helpful suggestion. We now use the simple carbon cycle box model used in Graven et al. (2020) and very similar to that of Keeling et al. (2017), to investigate the relative influence of changes in C_3 and C_4 plant abundance on $\delta^{13}\text{CO}_2$. The model has three biosphere boxes with fast, intermediate, and slow carbon turnover and associated low, intermediate, and high biomass, respectively.

We tested the model for different configurations based on our predictions of $\Delta^{13}\text{C}$, the relative fractions of C_3 and C_4 plants and differences in carbon use efficiency and turnover times between C_3 and C_4 plants. We assumed that box 1 with fast turnover/low biomass represents only C_4 photosynthesis, while the other two boxes represent only C_3 photosynthesis (one for C_3 herbaceous and the other one for C_3 woody). We then compared the observed and predicted $\delta^{13}\text{CO}_2$ for these configurations and discuss the differences with Keeling et al. (2017) findings.

I think it's clear that the P model is incompatible with the sensitivity from Keeling et al (2017) in the limit that the turnover time of C_4 is infinitely fast compared to C_3 . In this limit, the Keeling et al sensitivity of $0.014 \pm 0.007 \text{‰ ppm}^{-1}$ would apply to C_3 plants only. Although I'm a bit confused about the numbers (see below), the text suggests that the P model gets a smaller sensitivity for C_3 of 0.003‰ ppm^{-1} (line 163). The difference compared to the simple model of Keeling et al makes sense considering the differing assumptions about mesophyll effects. I think the paper needs to at least briefly address this limit and the incompatibility with Keeling et al in this limit.

We agree that our approach and that of Keeling et al. (2017) make different assumptions about mesophyll effects leading to different estimated trends in $\Delta^{13}\text{C}$ for C_3 plants. In a previously published study (Lavergne et al. 2022, GCB), we already discussed these differences. We are now referring to this study and briefly address in our discussion the discrepancy between the two approaches.

The authors consider cases with different turnover rates of C_3 and C_4 carbon, including a case which assumes the turnover time of C_4 is ten time shorter than that of C_3 ($\tau_{\text{model}} = 0.1$). Shouldn't this case be essentially equivalent to assuming that C_4 turnover is infinitely fast (i.e. the case in previous paragraph)? According to the text, however, this case yields a sensitivity of 0.018‰ ppm^{-1} (line 177), which is much larger than the value of 0.003‰ ppm^{-1} (line 163). This may be another indication that there is something fishy with the handling of the differential turnover of C_4 and C_3 .

We thank the reviewer for pointing this apparent inconsistency to us. The differences highlighted here are due to the different natures of the $\Delta^{13}\text{C}$ studied in the former manuscript draft (i.e. global land versus 'atmospheric' $\Delta^{13}\text{C}$ - calculated using Equations 4 and 6 from the former version, respectively).

The global land $\Delta^{13}\text{C}$ (including C_3 and C_4 plants) increases by 0.003 ‰ ppm^{-1} when the fraction of C_3 and C_4 plants is assumed to be constant. The increase of 0.018 ‰ ppm^{-1} in atmospheric $\Delta^{13}\text{C}$ to which the reviewer refers occurred when the relative fractions of C_3 and C_4 plants vary (i.e. the fraction of C_3 increases while that of C_4 decreases) but the turnover of C_4 is infinitely rapid. In this case, changes in $\Delta^{13}\text{C}$ for C_4 plants do not contribute significantly to the atmospheric $\Delta^{13}\text{C}$ but since the fraction of C_3 plants increases, 'atmospheric' $\Delta^{13}\text{C}$ increases more sharply than in the case where the C_3 fraction is constant. This explains the higher 'atmospheric' $\Delta^{13}\text{C}$ value observed (0.018 ‰ ppm^{-1}).

As mentioned above, and to avoid any confusion, we have modified our analyses and are now only reporting predictions of atmospheric $\delta^{13}\text{CO}_2$ (not anymore 'atmospheric' $\Delta^{13}\text{C}$ estimates). Since C_3 and C_4 biomasses are modulated by carbon use efficiency and turnover, we assume that variations in C_3/C_4 plant abundance have a smaller influence on land $\Delta^{13}\text{C}$ than on GPP. We now estimate the impact of variations in C_3/C_4 plant abundance on $\Delta^{13}\text{C}$ using information on the carbon use efficiency and turnover of the biome under consideration. As a result, the increase in $\Delta^{13}\text{C}$ is lower than previously reported, even when considering changes in C_3/C_4 abundance.

I suspect there may be a similar issue with the treatment of differential turnover with respect to soil carbon, but the method used is not very well described. I assume Eq (6) might have been applied here? Again, I think there's an issue because of the fishy mathematical limits, as I mentioned above. The formula here should be simple because one is addressing a steady state with just two time constants (no 40-year transient).

Since measured soil $\delta^{13}\text{C}$ is an average of the carbon isotopic signature of soil organic matter accumulated over several years, we estimated soil $\delta^{13}\text{C}$ from the model as the average of predicted $\delta^{13}\text{C}$ for both C_3 and C_4 plants, weighted by their relative annual abundance, over the entire period 1982–2016. We have now clarified this in the text.

I have trouble reconciling the trends shown in Figure 4e with numbers presented in the text. The green and purple curve(s) in Figure 4e, first panel (C_3) both increase by about 1 permil over 35 years, implying a rate of ~ 0.03 per mil per year. In contrast, the text (line 162) lists a rate of 0.005 permil per year, which is 6 times smaller. There would appear to be a similar problem the C_4 panel.

Perhaps a related issue is that the legend of Figure 4 is not clear. The main text implies Figure 4e is showing global averages for all C_3 or C_4 . But the caption indicates that the green curve is "Natural grasslands" and the violet curve is for "Natural and crops" (whatever that means). Neither curve would appear to be showing a combined measure over all C_3 or C_4 plants, as referenced in the main text.

The green and purple lines in Figure e (top panel) represent $\Delta^{13}\text{C}$ in C_3 plants for natural grasslands only and for grasslands and crops, respectively, estimated using our C_3 plant fraction (F_3) map determined using the C_4 fraction (F_4) estimated from our simple model as: $F_3 = 1 - F_4$. Figure e (bottom panel) is the same but for C_4 plants. In contrast, the brown line represents $\Delta^{13}\text{C}$ for C_3 or C_4 plants using the Still et al. (2009) map (which assumes a constant F_4 over time). In line 162, we have referred to the brown line, which is showing a moderate increase.

The purple lines in Figure 4e include both grasses and crops for C₄ plants and also woody plants for C₃ or and therefore represent the total $\Delta^{13}\text{C}$ for C₃ or C₄ plants. We agree that the legend may have been confusing. We have now edited it to clarify the figure.

On rereading, I note that the basis of the ratio F₄ is not defined. F₄ appears to be some fractional measure of cover. But what? Above ground biomass fraction? Fractional light interception? How would this be defined in the field?

We defined the fraction of C₄ plant (F_4) as the share of C₄ plants in the total gross primary production (GPP) or C₄ vegetation coverage. A similar definition has been used recently in Luo et al. (2024) Nature communications.

On rereading, I also note that $GPP_{C3,pot}$ and $GPP_{C4,pot}$ may not be defined.

In methods (L325) we mentioned that C₃, C₄ and total (C₃ + C₄) GPP were estimated from their respective potential GPP before introducing equation 3. We agree that we could have defined the terms more explicitly. This has now been done in the revised manuscript.

Reviewer #2 (Remarks to the Author):

This reviewer was satisfied with the responses from the authors, especially as related to the impacts of environmental forcing on trends in GPP and $\delta^{13}\text{C}$ discrimination. This reviewer thanks the authors for their detailed responses. I still have some lingering minor questions/suggestions which are listed below:

Has the model has been validated for the exact setup you are using in this manuscript including time period, spatial region, model parameterization and maybe most importantly; meteorological forcing? I realize the focus of this manuscript is how trends in C₃/C₄ transition impact $\delta^{13}\text{C}$, but trends in atmospheric forcing also play a role. The temporal trends in your meteorological forcing (Figure S5) show some conflicts (e.g. VPD and soil moisture). Although atmospheric CO₂ seems to be the main driver in the C₃/C₄ transition, it would also be interesting to show spatially explicit trends in VPD, temperature, and soil moisture to compare with trends in spatially explicit C₃/C₄ landcover. Figures S4 and S5 get close to this, but not exactly.

Cai et Prentice (2020) already validated the model for GPP predictions using eddy-covariance flux measurements. They used a similar meteorological forcing, i.e. temperature and VPD from CRU TS over 1982-2016, shortwave downwelling radiation from WATCH-WFDEI and greenness data from the GIMMS fAPAR 3 g dataset. They also ran the SPLASH model to estimate soil moisture. The only difference is that they used monthly CO₂ data from Mauna Loa Laboratory, while here we use yearly CO₂ data from Kohler et al. (2017). Given our focus on global annual rather than monthly changes in $\Delta^{13}\text{C}$, we decided to use a more spatially homogeneous CO₂ dataset to run our model. This choice should not affect the

validity of our model setup as the long-term trends in CO₂ concentrations in Mauna Loa Laboratory and Kohler et al. (2017) data are the same – only the mean values slightly differ.

Figure S4 shows the positive/negative influences of individual environmental variables on the predicted C₄ landcover across the globe, highlighting areas where, for example, an increase in VPD lead to significant increase in fraction of C₄ plants. It is a more powerful approach than visually comparing individual maps of spatial trends.

Nevertheless, we have modified Figure S5 to show spatial trends in VPD, temperature and soil moisture across the globe (instead of global averages) and briefly mentioned it in the Result section.

Figure 3a-3b: Given your legend and units and layout within this figure, it is almost impossible to visually detect any increasing C₄ trends (green) in these figures.

We have readjusted the legend and layout to make the increasing C₄ fraction trends more apparent.

Is the Figure S1 workflow correct? It claims the temperature forcing is coming from the WFDEI data product, however, the methods state the temperature is coming from CRU. The

methods state all met forcing data is derived from CRU with the exception of the SWdown which comes from WFDEI. This needs some clarification.

Apologize for the mistake! An earlier version of the workflow was mistakenly uploaded. We have now corrected the workflow.

“Figure S5 Global average temporal changes in (a) atmospheric CO₂ concentrations (ppm), (b) daytime air temperature (T_{air} , °C), (c) vapour pressure deficit (VPD, kPa) and (d) soil moisture (θ , m³ m⁻³)”

Define the meteorological forcing data (CRU and WRFDEI) here again for clarity. It is curious that the temperature trends are about steady whereas soil moisture is increasing and VPD is increasing. Soil moisture and VPD seem to show opposing trends, however, this may be because it is a seasonal and spatial average which hides some region specific behavior. Also could be because you are only showing photosynthetically active periods (as described in Methods) which is misleading when looking at decadal trends where general climate warming signal may not be fully captured. This should be clarified.

As mentioned previously, the meteorological forcing data for temperature and VPD comes from CRU TS, while the shortwave downwelling radiation comes from WATCH-WFDEI. We used the SPLASH model to estimate both PPF and soil moisture. The objective of our study is to determine whether recent changes in C₃/C₄ distribution and abundance influence significantly global land $\Delta^{13}C$ and ultimately atmospheric $\delta^{13}CO_2$, not to investigate annual decadal changes in hydroclimate.

In the former Figure S5 we show that both global VPD and soil moisture increase, although the increase is much more important in VPD. As the reviewer mentioned, the trends shown are focussed on photosynthetically active periods so are not representing the whole decadal trends. To avoid any confusion, we have now removed the figure. In the new Figure S5, we are now showing the temporal trend in daytime T_{air} , VPD and soil moisture across the globe, highlighting key hydroclimatic heterogeneity across the globe.

Figure S9 : What is the significance of the treecover > 58% threshold?

The treecover threshold was estimated based on the bimodal distribution between forest and non-forest grid cells as the minimum value between the two group of grid cells. The uncertainty in this value is $\pm 2\%$. We have clarified this in the text.

We thank the two reviewers for their valuable feedback on our revised manuscript. Please find below our response to their comments.

Reviewer #2 (Remarks to the Author):

The authors response to my remaining questions were acceptable.

We thank the reviewer for supporting the publication of our work.

Reviewer #3 (Remarks to the Author):

This manuscript focused on a scientific question that was argued in different references : Does the C3/C4 abundance influence the isotope composition of atmosphere? There exhibited an obvious innovation and this work will be benefit for an improvement of the carbon cycling under the background of global warming. However, there are some concerns for the method part.

We thank the reviewer for recognizing the value of our work for the field of carbon cycle research.

First, the manuscript classified the plants to C3 and C4, is it possible that the CAM plants disturbed the results?

It is unlikely that the exclusion of CAM plants in the analyses significantly influence the patterns we report for C₃ and C₄ plants. CAM species represent a relatively small fraction of global vegetation (Nobel 1991) and are concentrated in arid niches (Winter 2019), where their primary productivity is much lower compared to C₃ and C₄ plants (Nobel 1991). Our analysis focused on large-scale physiological responses to recent environmental changes, which are dominated by C₃ and C₄ vegetation (Luo et al. 2024). Moreover, CAM plants assimilate carbon dioxide (CO₂) primarily at night, following a distinct temporal pattern that makes their responses less directly comparable to daytime drivers of C₃ and C₄ photosynthesis such as light, temperature, and atmospheric CO₂.

Nobel, P.S. (1991). Achievable productivities of certain CAM plants: basis for high values compared with C3 and C4 plants. *New Phytologist*, 119, 183–205.

Winter, K. (2019). Ecophysiology of constitutive and facultative CAM photosynthesis. *Journal of Experimental Botany*, 70(22), 6495–6508.

Secondly, the estimation of isotope composition of soil was not well presented. For instance, in line 431, does this estimation consider the isotopic fractionation in soil process ?

We estimated soil carbon isotopic composition ($\delta^{13}\text{C}_{\text{soil}}$) as the weighted average of predicted $\delta^{13}\text{C}$ values for C₃ and C₄ plants, based on their relative annual fractions. We assumed that any additional isotopic fractionation within the soil was negligible over the study timeframe, and this clarification has been added to the Methods (L442-443).

Thirdly, a table describing the detail of remote sensing data including the resolution, duration and source was recommended.

We have now added Table S2 in Supplementary Material to provide more details on the climate and remote sensing data used in the study to run the C₃/C₄ model and simple carbon cycle model.

Fourth, although the manuscript suggested that the changes in C₃/C₄ did not fully explain the isotope compositions of the CO₂, the possible reasons should be described more specifically.

We have expanded on the potential causes of $\delta^{13}\text{CO}_2$ changes that may explain the mismatch between observations and simulations (L275-279).

'Uncertainties in biosphere processes—especially isotopic fractionation during post-photosynthetic pathways and soil respiration, and carbon residence times in soils—as well as ocean–atmosphere exchanges and fossil fuel emissions, can cause discrepancies between simulated and observed atmospheric $\delta^{13}\text{CO}_2$ ^{30,44}.'

Some specific comments are as follows:

(1) Line 38 : “weaker” is not a suitable word in the sentence.

We have edited the word as ‘smaller’.

(2) The link between GPP and delta C should better be described to enhance the logics in the introduction.

We have expanded on the link between GPP and $\Delta^{13}\text{C}$ in the introduction adding the following sentence in L39-42:

'While changes in GPP can influence carbon isotope discrimination ($\Delta^{13}\text{C}$) during photosynthesis, the relationship is not strictly linear and depends on environmental and physiological conditions², meaning shifts in plant productivity may affect $\delta^{13}\text{CO}_2$ in complex ways.'

(3) Some dataset like VPD and temperature may also produce some uncertainties. I recommended that you could emphasize the reliability of these products.

We used the CRU TS4.03 climate dataset, which provides monthly data on a 0.5° × 0.5° grid and is widely recognized in the climate community (Harris et al., 2020). From this dataset, we derived monthly mean daytime temperature (T_{daytime}) and vapor pressure deficit ($\text{VPD}_{\text{daytime}}$) to capture only the period when photosynthesis occurs. These variables were calculated using minimum and maximum temperatures (T_{min} , T_{max}) and actual vapor pressure (e_a) from CRU TS4.03, following the formula of Allen et al. (1998). Full methodological details are provided in Lavergne et al. (2020) and referenced in the main text. We consider the uncertainties introduced by calculating T_{daytime} and $\text{VPD}_{\text{daytime}}$ to be negligible compared with the broader uncertainties inherent in the original CRU TS4.03 dataset.

Allen RG, Pereira LS, Raes D, Smith M. 1998. Crop evapotranspiration - Guidelines for computing crop water requirements – FAO Irrigation and drainage paper 56. Irrigation and Drainage 1–15.

Harris, I., Osborn, T.J., Jones, P. et al. Version 4 of the CRU TS monthly high-resolution gridded multivariate climate dataset. Sci Data 7, 109 (2020). <https://doi.org/10.1038/s41597-020-0453-3>

Lavergne, A. et al. Historical changes in the stomatal limitation of photosynthesis: empirical support for an optimality principle. New Phytol 2484–2497 (2020) doi:10.1111/nph.16314.

(4) Actually, the isotope composition of CO₂, leaves, and soil all changed with seasons. Dose this variation influence your evaluation? Maybe a short discussion will be helpful for a better understanding of the seasonal effects.

Our $\delta^{13}\text{C}$ model estimates for CO₂, leaves, and soils are integrated across the growing season, thereby inherently accounting for seasonality. Because the study focuses on decadal, growing-season averaged changes in $\delta^{13}\text{C}$ rather than seasonal dynamics, we consider a discussion of seasonal effects beyond the scope of this work. As the main text already slightly exceeds the 5,000-word limit, we prefer to concentrate the discussion on the study's central findings.